# Visual stimulus features that elicit activity in object-vector cells

Sebastian O. Andersson [1✉], Edvard I. Moser [1] & May-Britt Moser [1✉]

Object-vector (OV) cells are cells in the medial entorhinal cortex (MEC) that track an animal's distance and direction to objects in the environment. Their firing fields are defined by vectorial relationships to free-standing 3-dimensional (3D) objects of a variety of identities and shapes. However, the natural world contains a panorama of objects, ranging from discrete 3D items to flat two-dimensional (2D) surfaces, and it remains unclear what are the most fundamental features of objects that drive vectorial responses. Here we address this question by systematically changing features of experimental objects. Using an algorithm that robustly identifies OV firing fields, we show that the cells respond to a variety of 2D surfaces, with visual contrast as the most basic visual feature to elicit neural responses. The findings suggest that OV cells use plain visual features as vectorial anchoring points, allowing vector-guided navigation to proceed in environments with few free-standing landmarks.

[1] Kavli Institute for Systems Neuroscience and Centre for Neural Computation, Norwegian University of Science and Technology (NTNU), Trondheim, Norway.
✉email: sebastian.o.andersson@ntnu.no; may-britt.moser@ntnu.no

Medial entorhinal cortex (MEC) and the adjacent pre- and parasubiculum are critical components of the neural representation of space[1,2], hosting cell types that dynamically signal the animal's position[3–5], head direction[5–7], speed[8] and proximity to borders[5,9,10]. These cells coexist and interact with a wide range of allocentrically tuned cell types in neighbouring hippocampal–parahippocampal regions, such as place cells[11,12], boundary-vector (BV) cells[13–15] and landmark-controlled cells[16–18], as well as various cells that encode position in egocentric coordinates relative to the animal's body axis[19–22]. Collectively, this network of space-coding neurons is thought to enable animals to navigate the world.

Recently, we reported the existence of yet another cell type in MEC: object-vector (OV) cells, which signal the animal's distance and direction to discrete objects in the environment[23]. The large number of OV cells in MEC, on par with the number of grid cells, suggests that tracking of the animal's position in an object-centred space is a major form of spatial representation in MEC[23], just as in the wider hippocampal formation[13–15]. OV cells respond to an impressive array of objects: from small to large, narrow to wide, and short to tall, although they are inclined to respond to taller objects[23]. In addition, the cells fire independently of the object's shape, colour or familiarity. While the stimulus space of objects that OV cells can produce vectors to must be gigantic, the lower bound of this stimulus space is unknown: what are the simplest object features that can elicit activity in OV cells? Here we address this question by parametrically stripping objects of some of their qualities while measuring responses of OV cells. We ask first whether OV cells respond only to free-standing three-dimensional (3D) objects by converting in a stepwise manner an object from 3D to two-dimensional (2D) and testing responses in OV cells. After finding that 2D surfaces on the wall of the recording environment are sufficient to elicit OV firing, we show that a simple visual contrast on the wall is sufficient to elevate activity.

## Results

**Identification of OV cells and OV fields**. To identify the most fundamental features that elicit activity in OV cells, we implanted seven mice ($n = 492$ cells) with tetrodes in the MEC (Supplementary Fig. 1), where OV cells are numerous[23]. As in earlier work[23], we performed three trials to identify OV cells: an 'Empty Box' trial, an 'Object' trial and a 'Moved Object' trial (Fig. 1a). In the latter pair of trials, the object was a $40 \times 8 \times 8$ cm (height × base area) multi-colour Duplo tower, known to produce clear OV responses[23] (Supplementary Fig. 2). OV cells were identified by an 'OV score', equivalent to the Pearson correlation between object-centred pairs of rate maps from the 'Object' and 'Moved Object' trials. These rate maps show firing rate (FR) as a function of allocentric angle and distance to the object (Fig. 1b). When the cell's firing depends on the animal's position relative to the object, and not on the animal's absolute position, this correlation should be high.

As in earlier work[23], cells had to pass multiple criteria to count as OV cells. First, we required the appearance of one or more new fields in the 'Object' trial compared to the 'Empty Box' trial. For each cell passing this criterion, we generated a shuffled spike-time distribution by randomly shifting the cell's spike timestamps along the animal's trajectory ($n = 200$ permutations per cell). Cells were defined as OV cells if (i) they had spatial information contents exceeding the 99th percentile of their own shuffling distribution (Fig. 1c), (ii) OV scores exceeding the 99th percentile of their own shuffling distribution (Fig. 1d), and (iii) the firing fields were positioned > 4 cm away from the object centre. In total, 67 out of 492 cells (13.6%) passed the triplet of criteria including the spatial information criterion (Fig. 1e) and the OV score criterion (Fig. 1f) and were classified as OV cells. The mean number of OV cells per mouse was 9.6 (minimum: 2 OV cells; maximum 14 OV cells) (Supplementary Table 1). The lowest fraction of OV cells across all mice was 5.7% while the highest fraction was 16.7%. The fraction of OV cells obtained overall (13.6%) was very similar to the fractioned obtained in the previous study recording OV cells in the same region (14.7%)[23].

Since the aim of the present study is to identify the basic features of objects that produce activity in OV cells, we expected some of the more primitive stimuli to elicit weak firing fields. To detect those weak fields, we assumed that a sensitive algorithm would be required. With this in mind, we developed a Bayesian algorithm that takes in spike data and computes a probability distribution for field locations. The algorithm models the data as coming from two sources: (1) a Gaussian that represents the cell's spatially specific firing field and (2) a uniform non-zero floor that represents the cell's tendency to produce noisy spikes anywhere in the animal's environment (Supplementary Fig. 3a). The algorithm uses an iterative procedure, evaluating each $(x, y)$ coordinate in the environment in 1 cm bins to determine which locations most likely host the cell's firing field. On simulated data where the ground truth position of the field was known, performance of the algorithm was nearly perfect across a wide range of noise levels for field sizes expected to be found in electrophysiological data (fields with 500 or 250 spikes had a mean error of 0 cm in a $100 \times 100$ cm$^2$ box; fields with 125 spikes had a mean error of 2.35 cm; the number of spikes drawn from noise was varied between 0, 500, 5000, 25,000, 50,000 and 100,000) (Supplementary Fig. 3c). On real data, the algorithm successfully identified OV cells with single (Supplementary Fig. 3d i, ii and iii) and multiple fields (Supplementary Fig. 3d iv).

**A 2D surface produces strong vectorial responses in OV cells**. Since previously we probed OV responses using discrete, 3D objects[23], we began by asking whether a flat 2D surface would be sufficient to generate vectorial responses. We used the original Duplo tower object (Supplementary Fig. 2b) and varied its volume by either gradually embedding it into the wall of the recording box or extending it out from the wall. Using a step-like procedure, we changed the object from a fully exposed 3D object, through a partly embedded object (50% standing out from the wall), to a flat 2D surface continuous with the box wall, or vice versa (Fig. 2a). In this experiment, as well as in further parametric experiments, we limited analysis to pre-selected OV cells, identified with a free-standing 3D object, using the criteria defined above. For these cells, we calculated two measures of responsiveness: (1) The FR inside a circular region of interest (ROI) in which we expected the cell to fire based on the cell's vector coordinates and (2) the cell's OV score[23]. For measure (1), the centre of the ROI was the location of the OV field identified with the Bayesian algorithm. This was taken as the maximum of the posterior probability distribution from the 'Object' trial originally used to identify the cell as an OV cell (Fig. 1a, middle; Fig. 2a, left: 'Reference trial'). The size of the ROI was always 15 cm, a pre-specified parameter. We confirmed that in all cases the results were independent of this parameter choice (Supplementary Fig. 4). For measure (1) we normalised the data for each cell, by expressing FRs as a fraction of the maximum value observed for each cell (1 = maximum, 0 = minimum). For measure (2), we used the initial 'Object' trial, where the standard Duplo tower was in the middle of the box, as a template (Fig. 1a, middle; Fig. 2a, left: 'Reference trial'). That is, measure (2) was the Pearson correlation between the object-centred rate map from the initial 'Object' trial (Fig. 1b, middle) and the object-centred rate map from one of the experimental trials.

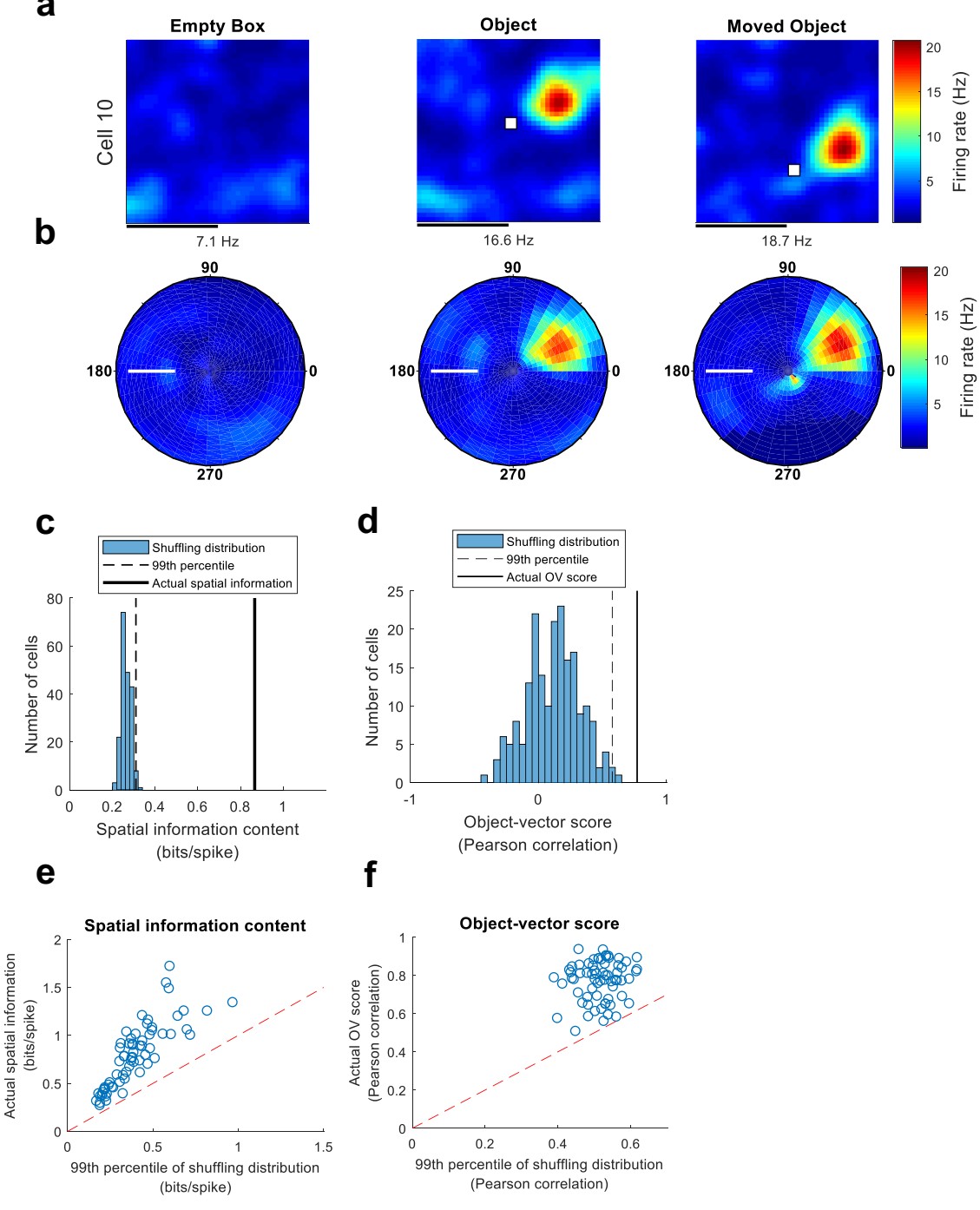

The presence of objects significantly influenced the OV cells' FR inside the ROI (Fig. 2c, left) ($H = 12144$, $p = 1.70 \times 10^{-6}$, $n = 30$ cells, Kruskal–Wallis test). All objects, including the plain 2D surface, produced an increase in FR inside the ROI relative to the Empty Box trial (2D vs. Empty Box, $W = 50$, $p = 2.92 \times 10^{-4}$; 50% vs. Empty Box, $W = 67$, $p = 0.0011$; 3D vs. Empty Box, $W = 58$, $5.63 \times 10^{-4}$; two-sided Wilcoxon signed-rank tests). The response to the 2D object was indistinguishable from the response both to the 3D object and the partially embedded object (2D vs. 3D, $W = 252$, $p = 0.4557$; 2D vs. 50%, $W = 256$, $p = 0.4051$, two-sided Wilcoxon signed-rank tests). There was no significant effect of the object on FR outside the ROI (Fig. 2c, right) ($H = 336$, $p = 0.7842$, $n = 30$ cells, Kruskal–Wallis test). The order of the change—from 3D to 2D, or vice versa—had no effect on the cells' FRs in the ROI

($n = 14$ experiments from 3D to 2D; $n = 27$ experiments from 2D to 3D; $W = 319$; $p = 0.1001$, two-sided Wilcoxon signed-rank test). The presence of objects also significantly influenced the cells' OV scores (Fig. 2d; Kruskal–Wallis test, $H = 0$, $p = 6.54 \times 10^{-8}$, $n = 30$ cells). All objects—3D or 2D—produced a significant increase in the OV score relative to baseline (2D vs. empty, $W = 94$, $p = 0.0044$; 50% vs. empty, $W = 35$, $p = 4.86 \times 10^{-5}$; 3D vs. Empty Box, $W = 30$, $p = 3.11 \times 10^{-5}$; two-sided Wilcoxon signed-rank tests). The OV score was larger for both the 3D object and the partially embedded object compared to the 2D object (2D vs. 3D, $W = 126$, $p = 3.07 \times 10^{-5}$, 2D vs. 50%, $W = 82$, $p = 0.0020$, two-sided Wilcoxon signed-rank tests).

To determine how generally the findings above apply to 2D surfaces, we next tested the cells' response to transparent 2D

**Fig. 1 Identification of object-vector (OV) cells. a** Rate maps for Example Cell 10 from the three trials used to identify it as an OV cell. In the first trial, the environment is empty ('Empty Box'). In the second trial, a free-standing object made of Duplo bricks is present in the environment ('Object'). In the third trial, the same object is moved to a new location ('Moved Object'). Rate maps show colour-coded firing rate in Hz as a function of the animal's position. White squares mark the object location. Peak firing rate (Hz) is indicated below each map. Scale bar represents 40 cm. The example cell fires in a specific distance and direction away from the object, the defining behaviour of OV cells. **b** Object-centred rate maps, displaying firing rate (Hz) as a function of the animal's distance (cm) and orientation (degrees) to the object. The object position is the centre of the map. The maps are from the 'Empty Box', 'Object' and 'Moved Object' trials for the cell shown in panel **a**. For an OV cell, we expect the maps from the 'Object' and 'Moved Object' trials to be similar. Scale bar in white, 20 cm. **c** Shuffling distribution of spatial information content for Example Cell 10 (same cell as in previous panels). The cell's spike timestamps were randomly shifted along the animal's trajectory ($n = 200$ permutations). For each shuffled cell, we calculated the spatial information content, which quantifies how informative the spikes are about the animal's position. The actual spatial information content of the cell is far above the 99th percentile of the shuffling distribution. Spatial information contents were calculated using data from the 'Object' trial. **d** Shuffling distribution of OV scores for Example Cell 10. After performing shuffling (as in the previous panel) we calculated the OV score for each shuffled cell. The OV score is the Pearson correlation between the object-centred maps shown in panel **b** from the 'Object' and 'Moved Object' trials. The actual OV score of the cell exceeds the 99th percentile of the shuffling distribution. **e** Scatterplot showing the actual spatial information content (bits/spike) of OV cells, compared to the threshold value obtained from each cell's shuffling distribution. The threshold was the 99th percentile of the shuffling distribution. All data points fall above the diagonal because, by definition, OV cells need to pass the spatial information criterion. The spatial information content was calculated on the 'Object' trial (**a**). **f** Scatterplot showing the actual OV score of OV cells, compared to the threshold value obtained from each cell's shuffling distribution. The threshold was the 99th percentile of the shuffling distribution. All data points fall above the diagonal because, by definition, OV cells need to pass the OV score criterion. The OV score is the Pearson correlation between the object-centred maps in (**b**) from the 'Object' and 'Moved Object' trials.

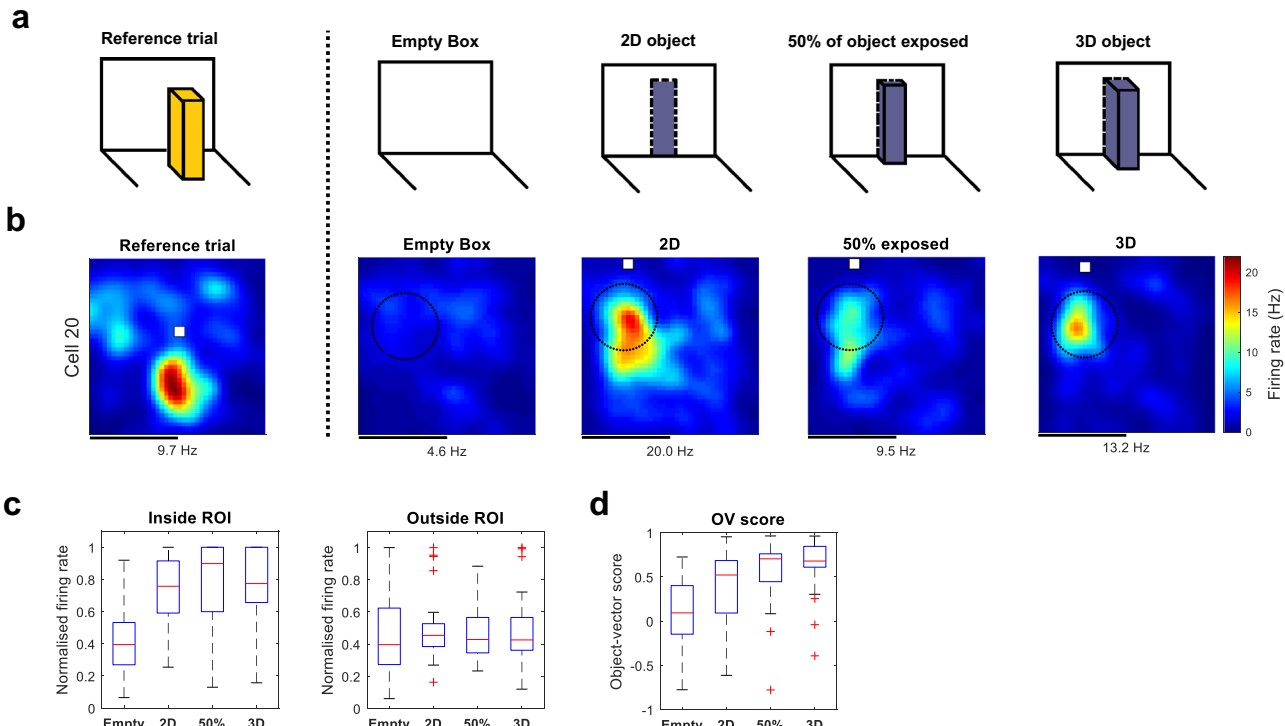

**Fig. 2 Object-vector cells respond to two-dimensional objects. a** Experimental design with four different trials: mice foraged in an environment with either an object absent ('Empty Box') or present ('2D', '50%', '3D'). In the three object trials, we varied the amount of the object's volume exposed to the animal. In the 2D trial, the visible part of the object was a flat 2D surface, with the rest completely embedded into the wall. In the middle trial, the object was partially embedded into the wall so that 50% of its volume was exposed. In the 3D trial, the full volume of the object was exposed. The reference trial, used to find the cell's vector coordinates, is shown on the left. This was the original 'Object' trial used to identify the cell as an OV cell (Fig. 1a, middle). **b** Colour-coded rate maps from example OV cell that responded as strongly to 2D surfaces as to 3D objects. Rate maps show colour-coded firing rate in Hz as a function of the animal's position. The white square marks the object location. The dotted circle marks the ROI in which we expected the cell to fire based on its vector coordinates. The vector coordinates of the cell were found by applying the algorithm from Supplementary Fig. 3 to the 'Reference trial' on the left. This trial is also what we used as a template for computing the OV score. Peak firing rate (Hz) is indicated below each map. Scale bar, 40 cm. **c** Box-and-whisker plots of normalised firing rates (Hz) of object-vector cells ($n = 30$) as a function of object dimensionality. Firing rates were normalised to the maximum data point across all eight data points observed for each cell (4 experimental conditions × inside/outside ROI) so that the maximum rate is 1. The middle line (in red) on each box indicates the median, while the bottom and top lines (in blue) indicate the lower and upper quartiles, respectively. Whiskers show the range of data in each condition. Red crosses show outliers that lie more than 1.5 times outside the interquartile range. The firing rate was measured either inside (left) or outside the ROI (right). **d** Box-and-whisker plot of object-vector scores (ranging from −1 to 1) as a function of object dimensionality. The object-vector score is equivalent to the Pearson correlation between pairs of object-centred rate maps (see 'Method' for details). Here, the reference rate map is from the 'Object' trial originally used to identify the cell.

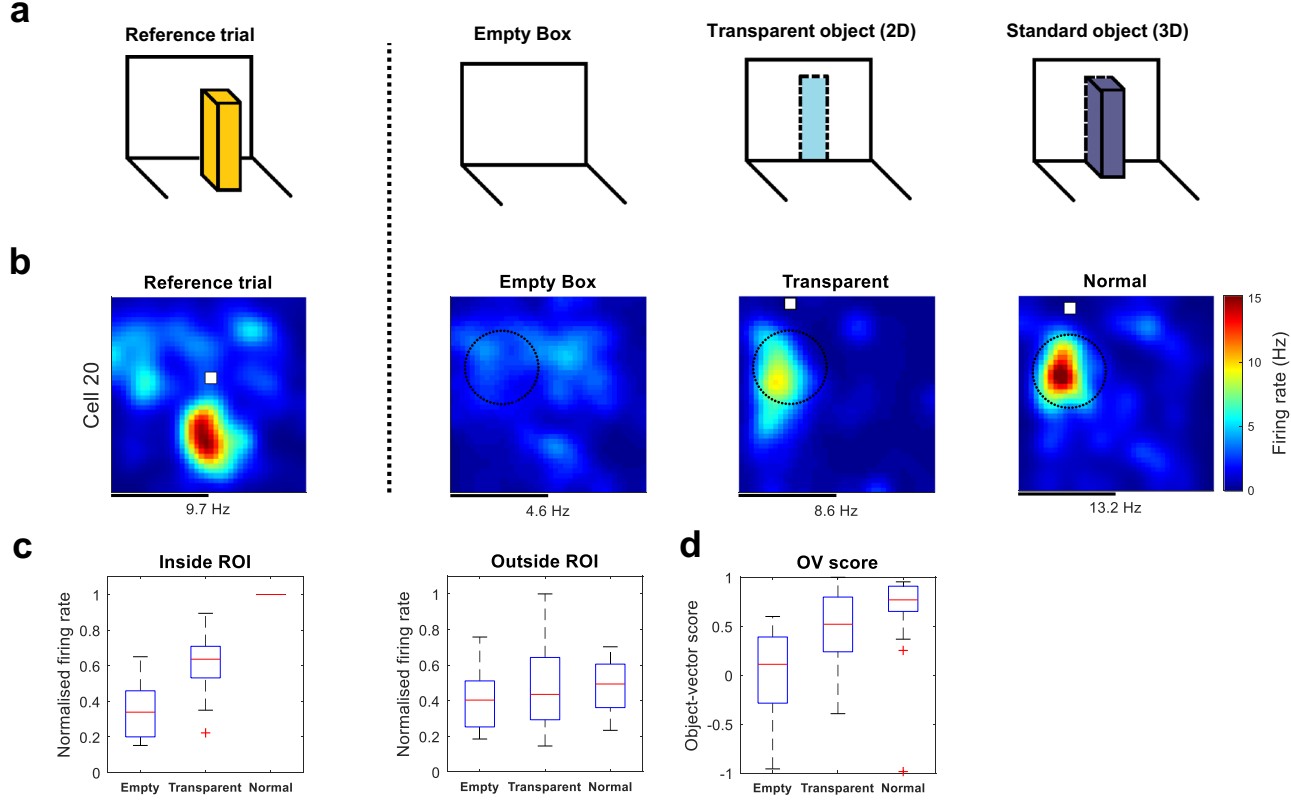

**Fig. 3 Object-vector cells respond to transparent objects. a** Experimental design with three different trials: either the box was empty, had a transparent object present, or a standard object present (a Duplo tower). The reference trial, used to find the cell's vector coordinates, is shown on the left. **b** Rate maps from example OV cell that responded strongly to the transparent object. Rate map conventions are as in Fig. 2. Scale bar, 40 cm. **c** Box-and-whisker plots of normalised firing rates (Hz) of OV cells ($n = 14$) as a function of discreteness of the object, inside (left) or outside the ROI (right). **d** OV score as a function of object type. Conventions for box-and-whisker plots are as in Fig. 2.

surfaces (Fig. 3a). In this experiment, the 'object' was a transparent film covering a hole in the box wall, with the hole cut to the same size as the Duplo tower ($40 \times 8$ cm) (Supplementary Fig. 2c). OV cells responded robustly to this object as well (Fig. 3c, left; Transparent vs. Empty Box, $W = 9, P = 0.0081$, two-sided Wilcoxon signed-rank test), although less strongly than to the 3D Duplo tower (3D vs. Transparent, $W = 0, p = 2.44 \times 10^{-4}$, two-sided Wilcoxon signed-rank test). As before, the effect on firing was not present outside the ROI (Fig. 3c, right) ($H = 0$, $p = 0.4214, n = 14$ cells, Kruskal–Wallis test). In addition, the transparent object increased the OV score (Transparent vs. Empty Box, $W = 0, p = 0.0166$, two-sided Wilcoxon signed-rank test) to a level indistinguishable from the 3D object (Fig. 3d) (Transparent vs. 3D, $W = 26, p = 0.1040$, two-sided Wilcoxon signed-rank test). Taken together, the results demonstrate that OV cells respond to a variety of flat 2D surfaces, often as strongly as they respond to discrete 3D objects.

**A visual contrast on the wall is sufficient to elicit vectorial responses in OV cells.** Having shown that OV cells can use 2D surfaces as reference points, we next asked whether the surface could be stripped down further to isolate the underlying features that prompt OV cells to respond. Given that vision likely has an important role in OV firing (see 'Discussion') we wondered whether a visual contrast—the most primitive kind of 2D object in the present study—could be sufficient to generate vectorial responses. To probe responses to visual contrast, we used a band of self-adhesive tape printed on the wall of the recording chamber (Supplementary Fig. 2d). Again, in a step-like procedure, we

changed the contrast from 0% (no self-adhesive tape), through 10% and 60%, to 100% whiteness ('Empty', 'Dark Grey', 'Grey' and 'White', respectively) (Fig. 4a). While some cells had FRs inside the ROI that clearly increased with visual contrast (Fig. 4b) other cells were more difficult to evaluate (Supplementary Fig. 5a), suggesting that we had approached the lower bound of the cells' preferred stimuli. Looking at the sign of the change in FR inside the ROI, and pooling across all visual contrasts, OV cells responded positively in 34 instances and failed to respond in 8 instances.

There was a significant effect of contrast on the FR of OV cells inside the ROI (Fig. 4c, left) ($H = 720, p = 0.0177, n = 14$ cells, Kruskal–Wallis test). OV cells showed a significant response to the white contrast ($W = 6, P = 0.0034$) but only a non-significant trend to a response to the grey contrast ($W = 19, P = 0.0803$) and the dark grey contrast ($W = 23, P = 0.1272$). No effect of contrast was present outside the ROI (Fig. 4c, right) ($H = 120$, $p = 0.6310, n = 14$ cells, Kruskal–Wallis test). While contrast did not have a statistically significant effect on the OV score (Fig. 4d) ($H = 0, p = 0.4112, n = 14$ cells, Kruskal–Wallis test) we note that, for 2D objects, the OV score suffers from a binning problem since the animal can only explore half of all allocentric angles (180° out of the full 360°). This makes the FR inside the ROI a more sensitive measure for these types of features. Taken together, our results establish that a simple visual feature such as contrast modulates the FR of OV cells.

**The probability that OV cells respond to different features and objects.** The post hoc statistical tests used above, while

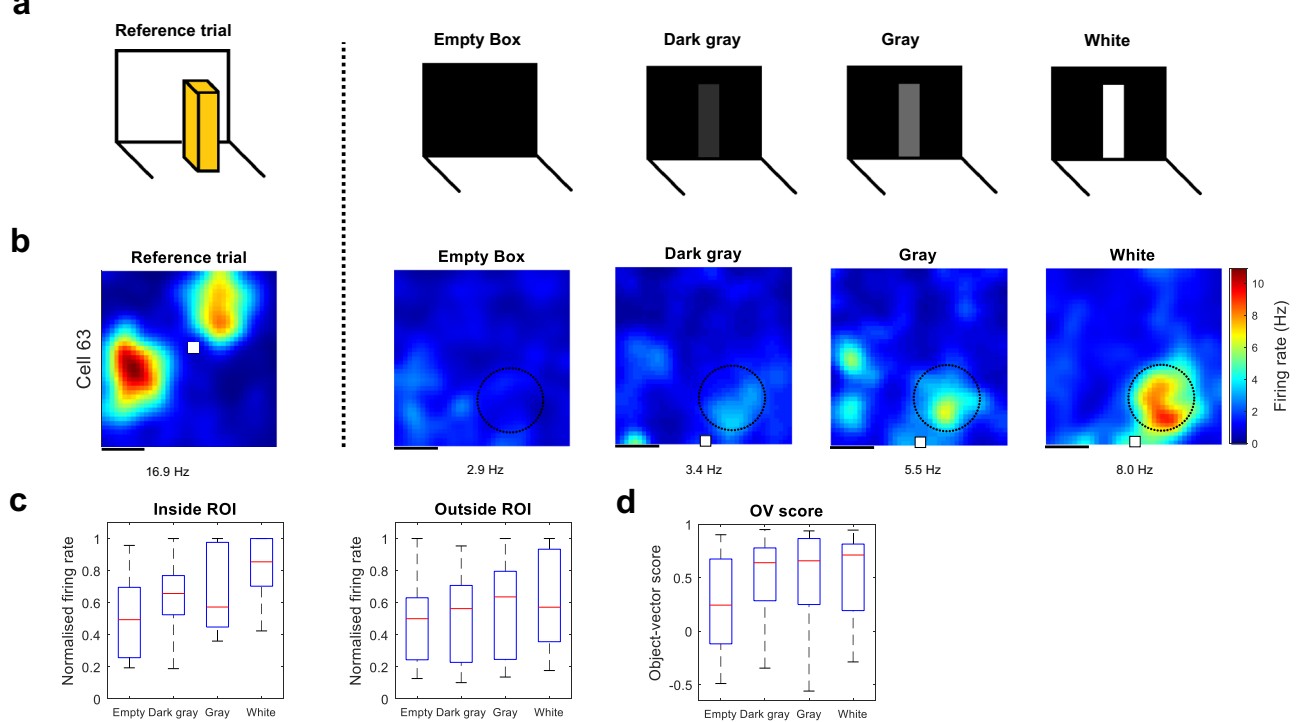

**Fig. 4 Responses of object-vector cells to visual contrasts. a** Experimental design with four different trials: either the box was empty or had a visual contrast present. In the three trials with a visual contrast, we varied the object's contrast from dark grey, to grey, to white (10%, 60% and 100% whiteness, respectively). The reference trial, used to find the cell's vector coordinates, is shown on the left. **b** Example rate maps from OV cell clearly increasing its firing rate as a function of visual contrast. Rate map conventions are as in Fig. 2. Scale bar, 20 cm. **c** Box-and-whisker plots of normalised firing rates (Hz) of OV cells ($n = 14$) as a function of the object's visual contrast. The firing rate was measured either inside (left) or outside the ROI (right). **d** Box-and-whisker plot of OV scores as a function of visual contrast. Conventions for box-and-whisker plots are as in Fig. 2.

advantageous in many respects, have limitations when it comes to comparing responses to different visual contrasts. For example, the response to the white contrast was found significant and the response to the grey contrast insignificant, but this makes it difficult to compare: (i) how much more likely is it that OV cells respond to the white contrast, compared to the grey contrast? (ii) how much higher is the likely range of FR values for the white contrast, compared to the grey contrast? For this reason, we conducted a complementary statistical analysis that quantifies for every visual contrast (i) the probability that OV cells respond; (ii) a credible region that expresses 'there is a 95% probability that OV cells respond with an average FR between [$a =$ lower bound] and [$b =$ upper bound]'. Measures (i) and (ii) can be readily calculated with a Bayesian statistical analysis, a method based on classical work[24–26] and well-advanced in many fields[27–29]. Given the subtle responses in the visual contrast experiment (Fig. 4), the analysis is especially informative here, but we performed the analysis for all experiments of the present study (Fig. 5).

To perform the analysis, we calculated the probability distribution $P(\mathrm{FR}|D)$, the probability that OV cells respond with a particular average FR to some object (for example, the white contrast) given the data collected ($D$). By Bayes' theorem, this is given by

$$P(\mathrm{FR}|D) \propto P(D|\mathrm{FR})P(\mathrm{FR}) \qquad (1)$$

where $P(D|\mathrm{FR})$ is the likelihood and $P(\mathrm{FR})$ is the prior (Fig. 5a). We assigned a Gaussian distribution for the likelihood and a uniform distribution for the prior (see 'Methods' for details). To develop some intuition for the method, note that if the data clustered around 2.8 Hz, the dataset would be likely to have been

produced by a Gaussian with mean at 3 Hz, but unlikely to have been produced by a Gaussian with mean at 20 Hz. Using the notation above, $P(D|\mathrm{FR} = 3\,\mathrm{Hz}) \gg P(D|\mathrm{FR} = 20\,\mathrm{Hz})$. The main idea behind the method is to systematically move the Gaussian across the real number line (e.g. from FR changes of $-10$ to $10$ Hz in steps of 0.01 Hz) in order to find which mean values are most consistent with the dataset obtained. This gives a probability distribution for different levels of responding to each object (Fig. 5b–d). More specifically, the data (the FR inside each OV cell's ROI) are expressed relative to the 'Empty Box' session,

$$\text{FR inside ROI in experimental condition} - \text{FR inside ROI in 'Empty Box' condition}$$

$$(2)$$

which means that in the probability distributions, probability mass on the right (left) of 0 Hz should be interpreted as a positive (negative) response to the object.

The results from this analysis are summarised in the probability distributions (Fig. 5b–d), showing the probabilities of different average FRs, for each specific object or feature. For the objects in the first set of experiments, where we varied the discreteness of the object, all curves were in the positive range, occupying locations to the right of 0 Hz (Fig. 5b). Thus, the probability that OV cells respond with a positive change in FR was 1 ('3D'), 1 ('50%') and 1 ('2D'), respectively (Fig. 5b; 95% probability that response was between [1.12 Hz, 2.62 Hz], [1.22 Hz, 3.06 Hz] and [1.35 Hz, 2.96 Hz]). Consistent with OV cells responding as strongly to 2D surfaces as 3D objects, the curves were overlapping (Fig. 5b). Similarly, the probability that OV cells respond to the transparent object was 0.9987 and 1 for the 3D tower in the same experiment (Fig. 5c). However, the

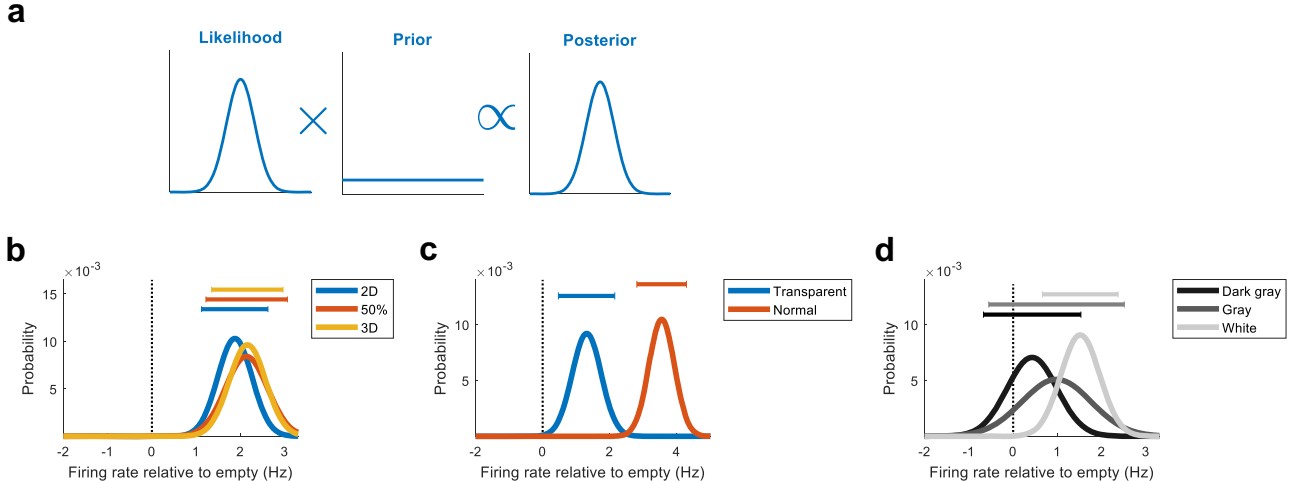

**Fig. 5 Posterior probability distributions of the average firing rates of OV cells to each object or feature. a** Visualisation of Bayesian inference. Bayes' theorem multiplies the likelihood with a prior to give the posterior distribution summarising our state of knowledge. The likelihood shows which average firing rates (FRs) best explain the data obtained. The prior gives the probabilities of different firing rates before seeing the data. The resulting posterior shows the probability that OV cells respond with any average FR to the object. For the distributions in Fig. 5, we assigned a Gaussian likelihood and assumed a uniform prior (see 'Methods' for details). We verified that our conclusions were insensitive to these assumptions (Supplementary Figs. 7 and 8). **b** Posterior distributions showing the probability that OV cells respond with any average FR to each object. The plots correspond to Fig. 2 and describe the results for the 2D surface (blue curve), the partially embedded object (red curve) and the 3D object (yellow curve). The data are the firing rates of OV cells inside the region in which we expect each cell to fire based on its vector coordinates. The firing rate in the same region in the 'Empty Box' trial has been subtracted. This means that probability on the right (left) of 0 Hz should be interpreted as a positive (negative) response to the object. 0 Hz is marked by the stippled line. The intervals at the top represent the 'credible region', which is the smallest possible region containing 95% of the probability mass. The credible region can be interpreted as "there is a 95% probability that OV cells respond with an average FR between these bounds'. **c** Same as in the previous panel but probability distributions corresponding to Fig. 3 describing the results for the transparent surface (blue curve) and normal 3D tower (red curve). **d** Same as in the previous panels but probability distributions corresponding to Fig. 4 describing the results for the dark grey, grey and white contrast. Note that while all curves peak at positive values, the right-shift of the curves increases gradually with increasing visual contrast.

strength of the response was lower for the transparent object (with 95% probability between 0.47 and 2.15 Hz) than the normal 3D object (with 95% probability between 2.81 and 4.29 Hz). The separation between the curves makes this difference in responding evident (Fig. 5c).

Visual contrast had an incremental effect on the probability distributions (Fig. 5d). For the dark grey contrast, only a moderate amount of probability (0.77) was above 0 Hz. For the grey contrast, a larger amount of probability (0.89) was above 0. For the white contrast, nearly all probability mass (0.9997) was in the positive range. That is, the probability that OV cells respond to the white contrast is nearly 1. The trend is seen visually as a right-shift of the curves (Fig. 5d). A similar trend was present when applying the same analysis to the OV score (Supplementary Fig. 6a–c). We found that, with 95% probability, OV cells respond to the white contrast with an average FR between 0.66 and 2,37 Hz. The corresponding ranges for the grey and dark grey contrasts were [−0.55 Hz, 2.51 Hz] and [−0.67 Hz, 1.53 Hz], respectively. Taken together, these results show that OV cells are likely to respond to the grey contrast, but almost certainly respond to the white contrast. This establishes that a visual contrast is sufficient for OV cells to respond, with increasing certainty for higher contrasts. The results were insensitive to the choice of prior (Supplementary Fig. 7) and likelihood function (Supplementary Fig. 8).

## Discussion
In this study, we have identified some of the simplest types of objects and features that make up the stimulus space of OV cells.

Using a parametric approach, we show that the cells' capability to encode vectors to discrete objects extends to 2D, as well as transparent, visual stimuli. In the most extreme case, a simple visual contrast was sufficient to elicit OV firing. The OV response increased gradually as we increased the contrast of the object. Since low-contrast features either failed to elicit a response, or only elicited a weak response, while high contrast features consistently succeeded, visual contrast changes are probably among the minimum and fundamental stimulus changes that excite OV cells. In the absence of a population of cells where some cells respond to low-contrast stimuli and others do not, we cannot rule out, however, the possibility that moments of inattention contribute to the weak response to these stimuli in many cells.

The results show that vision is sufficient to drive responses in OV cells. Combined with our previous finding of impaired OV firing in darkness[23], this shows that vision is both necessary and sufficient to prompt OV responses. However, the results do not rule out that other types of sensory stimuli—such as tactile or olfactory stimuli—also could elevate OV activity. A residual OV response is present even in complete darkness[23], which might result from tactile or olfactory information. Similarly, the amount of tactile information for the visual contrasts was minimised but not absent (for example, a faint edge between the wall and the visual contrast was present). Consequently, while we have identified a fundamental feature that drives OV activity, future work will have to determine whether other sensory features of objects, such as tactile or olfactory cues, also could elicit vectorial responses.

The proposal that vision has a fundamental role in driving OV activity might set them apart from border cells[5,9,10]. For border

cells, blockade of the animal's trajectory may be necessary since the cells do not respond to elevated borders[23]. Interestingly, BV cells in the subiculum do respond to gaps in the environment that the animal can traverse[14,15], raising the possibility that the underlying mechanism generating firing in border cells and BV cells is different[23]. While a fundamental stimulus feature for firing in OV cells is identified, it is still unknown what the equivalent minimal features for border cells and BV cells are.

A further striking difference between these functional cell types is the spectrum of distances they encode. In OV[23] and BV cells[13–15] firing fields cover both proximal and distal locations, while border cells have only proximal firing fields[5,9,10]. This means that OV and BV cells can represent the animal's position relative to discrete objects and features (in the case of OV cells) and relative to boundaries (in the case of BV cells) whether these items are near or far. Border cells represent nearby boundaries that obstruct the animal's immediate path[23]. The distinct natures of these cell types—in terms of (1) distance tuning and (2) object selectivity—provide a complementary and rich representation of the environmental layout, regardless of which specific objects, features and boundaries are present in the environment. Despite their differences, the computational goal of supporting landmark-based navigation is likely shared between OV cells, BV cells and border cells[15].

The finding that a visual contrast alone is sufficient for OV cells to respond suggests that even when no salient object is present in the environment (such as in the 'Empty Box' trial) OV cells might respond to corners, edges, shadows and other types of contrast. Consistent with this, OV cells are rarely silent in the 'Empty Box' trial but often show one or more firing fields (Supplementary Fig. 5b, c)—usually weaker than the object-induced field. Without other knowledge it would be easy to disregard this firing as neuronal 'noise'[30]. However, the present results suggest that this firing might not be 'noise' but a real effort by OV cells at signalling the presence of the few landmarks available in a nearly content-free environment.

What implications does this have for spatial representation in MEC in general? Given the substantial number of OV cells[23] and given the present results, researchers that record from MEC in the absence of discrete free-standing objects could expect to find spatial firing fields simply because of the sensitivity of OV cells to visual contrasts. These firing fields would be (1) spatial, if the cues remain in the same place; and (2) non-periodic, because the cues are not periodic. From this perspective, our findings may explain the presence of 'aperiodic spatial cells' in MEC recorded by us and others[31–35]: these are cells with significant spatial or positional information that are not grid cells. In one study, the fraction of such MEC cells was estimated at 68%[33]. The present results raise the possibility that a fraction of these aperiodic spatial cells are OV cells responding to visual contrasts such as corners or shades within an otherwise stimulus-deprived environment.

The proposal that aperiodic spatial cells could be OV cells has one more implication. We have previously reported that aperiodic spatial cells depend on activity of somatostatin (SOM) interneurons while grid cells depend on parvalbumin (PV) interneurons[35]. Now, if the proposal that aperiodic spatial cells are OV cells is correct, this would imply (1) that OV cells require activity of SOM interneurons and (2) that OV cells and grid cells are modulated by distinct subclasses of inhibitory interneurons. While this has not yet been tested, distinct inhibitory control of OV cells and grid cells might suggest that the two cell types separate into two distinct systems: one involving grid cells (dependent on PV interneurons) for representing the animal's position by updating self-motion[2,36] and one involving OV cells (putatively dependent on SOM interneurons) for vectorial

representation of the animal's position by anchoring onto objects and their visual features.

## Methods

**Subjects.** Data were obtained from seven wild-type C57/BL6 mice (six males, one female) aged 3–12 months. All mice were kept on a 12 h light/12 h dark schedule in a humidity and temperature-controlled environment. The mice were housed in single mouse cages after implantation. Experiments were performed in the dark phase. The mice were not deprived of food or water. Experiments were performed in accordance with the Norwegian Animal Welfare Act and the European Convention for the Protection of Vertebrate Animals used for Experimental and Other Scientific Purposes.

**Surgery and electrode implantation.** The mice were anaesthetised with 5% isoflurane (air flow: 1.2 l/min) in an induction chamber. After induction of anaesthesia, the mice received subcutaneous injections of (a) buprenorphine (Temgesic, 0.03 mg/ml solution and 0.05 mg/kg of animal's body weight), a centrally acting opioid and (b) meloxicam (Metacam, 1 mg/ml solution and 5 mg/kg of animal's body weight), a non-steroidal anti-inflammatory drug that inhibits prostaglandin synthesis. The purpose of using these drugs is to reduce postoperative pain (buprenorphine, meloxicam) and to reduce inflammation (meloxicam). The mice were then fixed in a Kopf stereotaxic frame for implantation. The local anaesthetic bupivacaine (Marcain, 0.5 mg/ml solution and 1 mg/kg of animal's body weight) was injected subcutaneously before the incision was made, in order to provide pain relief at the site of the incision. During surgery, isoflurane was gradually reduced from 3 to 1% according to the animal's physiological condition. The depth of anaesthesia was monitored by testing tail and pinch reflexes as well as breathing. Anaesthetised mice were implanted with a single bundle of four tetrodes attached to a microdrive fastened to the skull of the mouse. The tetrodes were targeted to MEC at an angle of 3° relative to the bregma/lambda horizontal reference plane, with tips pointing in the posterior direction. The tetrodes were inserted 3.2–3.3 mm lateral to the midline and 0.4 mm anterior to the transverse sinus edge, with an initial depth of 900 µm. The implant was secured to the skull with histoacryl and dental cement. One screw was connected to the drive ground. Tetrodes were constructed from four twisted 17-µm polyimide-coated platinum-iridium (90–10%) wires (California Fine Wire). The electrode tips were plated with platinum to reduce electrode impedances to between 120 and 220 kΩ at 1 kHz.

**Recording procedure.** Data collection started 1–2 weeks after implantation of tetrodes. During recording, the animal was connected to an Axona data acquisition system (Axona) via an AC-coupled unity-gain operational amplifier close to the animal's head, using a lightweight counterbalanced multiwire cable from both implants to an amplifier. Unit activity was amplified 3000–14,000 times and band-pass filtered between 0.8 and 6.7 kHz. Triggered spikes were stored to disk at 48 kHz with a 32-bit time stamp. An overhead camera recorded the position of two light-emitting diodes (LEDs) on the head stage, each at a sampling rate of 50 Hz. The diodes were separated by 3 cm. To sample activity at multiple dorsoventral locations, the tetrodes were lowered in steps of 25–50 µm. Recordings started when the tetrodes were judged to be in the MEC, using theta modulation, presence of spatial and directional cell types and as well as recording depth as criteria.

**Spike sorting and cell classification.** Spike sorting was performed offline using graphical cluster-cutting software (ctools, T. Waaga). Spikes were clustered manually in 2D projections of the multidimensional parameter space (consisting of waveform amplitudes), using autocorrelation and cross-correlation functions as additional separation tools and separation criteria. Cluster separation was assessed by calculating distances between spikes of different cells in Mahalonobis space[37] (median isolation distance: 16.3; 25th and 75th percentiles: 11.1–26.8). Noise in the vicinity of clusters was expressed as the L ratio[37] (median L ratio: 0.18; 25th and 75th percentiles: 0.05–0.78). Clusters on successive recording sessions were identified as the same unit if the locations of the spike clusters were stable.

**Behavioural procedures.** The mice were trained to forage for cookie crumbs in an 80 cm × 80 cm square arena, enclosed by 50-cm-high walls. Cookie crumbs were thrown out one-by-one in random locations. Thick dark blue curtains surrounded the recording arena, except for an opening on one side. Before testing, the mouse rested outside the curtain in a plexiglass cage coated with towels. Testing was performed at low light levels to encourage exploration. Between trials, the mat covering the floor of the recording box was cleaned.

*Screening for OV cells.* To screen for OV cells we first performed an 'Empty Box' trial in which no object was present in the arena, followed by an 'Object' trial in which a tower-shaped Duplo object (Supplementary Fig. 2a) was placed in the centre of the arena. In OV cells, the 'Object' trial was expected to induce new firing compared to the 'Empty Box' trial. To verify that firing was in a specific distance and direction away from the object, we performed a third 'Moved Object' trial in which the object was moved in a pseudo-random fashion to a new location. Trials were typically spaced by a few minutes, during which the experimenter clustered

and inspected recorded cells. The identity of the object used for screening trials was the same from day to day. Most trials were 30 min long (mean recording time 29.3 min).

*2D/3D experiment*. To determine whether OV cells produce vectorial responses to flat 2D surfaces, and whether they are comparable to responses to the discrete 3D objects used in previous work[23], we used a step-like experimental protocol in which the discreteness of an object was varied (Fig. 2). In the first trial, a tower-like Duplo object ($40 \times 8 \times 8$ cm) was completely embedded into the wall of the recording arena through an opening with the same height and length as the object ($40 \times 8$ cm) (Supplementary Fig. 2b). Thus, in this trial, the object was a flat 2D surface continuous with the wall ('2D'). In the second trial, the same object was only partially embedded so that half of its volume was exposed to the animal ('50%'). In the third trial, the entire volume of the object was exposed to the animal ('3D'). The width of object extending into the arena was 0, 4 and 8 cm, respectively. Most experiments ($n = 27$) were performed in the direction from 2D to 3D as above, but we also performed experiments ($n = 14$) in the opposite direction, from 3D to 2D. Most trials were 30 min long (mean recording time 30 min).

*Transparent-object experiment*. To verify the findings from the 2D/3D experiment and confirm that a variety of flat 2D surfaces can induce vectorial responses in OV cells, we performed an additional experiment with transparent 2D objects (Fig. 3). The experiment began with a reference 'Empty Box' trial without an object present. In the second trial, an opening in the arena wall (height and length $40 \times 8$ cm$^2$) was covered with transparent film to produce an opaque 2D surface (Supplementary Fig. 2c). The behaviour of the mice suggested that they noticed the presence of this 'object'. During the trial, dark blue curtains surrounded the arena (curtains located about 1 m away from the arena walls), preventing the animal from seeing any other material or object than the uniform curtain behind the transparent surface. In the third trial, a tower-like Duplo object was placed in the arena, in the same location where the transparent object had been. The purpose of the last trial was to compare any firing to the transparent object to firing to a standard object known to elicit strong vectorial responses. Most trials were 30 min long (mean recording time 30 min).

*Contrast experiment*. Having shown that 2D surfaces are sufficient to elicit vectorial responses, we asked whether the fundamental stimuli that elicit activity in OV cells are visual contrasts (Fig. 4). The experiment began with an 'Empty Box' trial with no visual contrast present. In the next three trials ('Dark Grey', 'Grey' and 'White', respectively) the degree of whiteness of a visual contrast on the arena wall was varied between 0%, 10%, 60% and 100%. The visual contrast was a $40 \times 8$ cm$^2$ band of self-adhesive tape printed on the arena wall (Supplementary Fig. 2d). The printed material had the same texture as the rest of the arena wall. The arena wall on which the contrast was put (N, S, E or W wall) varied depending on the OV cell's properties. For example, if the OV cell had a firing field north of the object, the visual contrast would be placed on the south wall. Changes in location for the contrast cue were made by rearranging the locations of the entire walls. Most trials were 30 min (mean recording time 28.2 min).

*Quantification of behaviour*. We also quantified how mice's behaviour differed for different object types (Supplementary Fig. 9). Firstly, we calculated the fraction of time that the animal spent near the object (defined as position samples in which the animal's distance was < 25 cm from the object) (Supplementary Fig. 9a). Secondly, we calculated the mean distance of the animal to the object (the average being taken over a single trial). Both these measures are proxies for the animal's interest in the object. Within the different experiments, animals showed similar levels of interest for the different object types. In the 2D/3D experiment, the median fraction of time near the object was 0.21 (2D), 0.23 (partly embedded and 0.27 (3D). The median average distance was 44.4 cm (2D), 44.2 cm (partly embedded) and 43.5 cm (3D). In the transparent-object experiment, the median fraction of time near the object was 0.28 (transparent) and 0.30 (standard). The median average distance was 41.6 cm (transparent) and 44.0 cm (standard). Animals spent less time near the visual contrasts compared to other objects. However, for different visual contrasts, the values were similar. The median fraction time near the object was 0.15 (dark grey), 0.13 (grey) and 0.13 (white). The average distance to the object was 46.1 (dark grey), 46.6 (grey) and 46.6 (white). Finally, we calculated the percentage of the environment explored by the animal, defined as the percentage of $2 \times 2$ cm spatial bins that the animal covered (Supplementary Fig. 9c). The coverage was high for all object types (median always > 92%).

**Firing rate maps**. To produce FR maps (Fig. 1a), position estimates were convolved with a 35-point Gaussian filter and *x*, *y* coordinates were sorted into $2$ cm $\times 2$ cm bins. Spike timestamps were matched with position timestamps. Firing rate maps were determined by counting the number of spikes falling in each bin and dividing by the amount of time spent in that bin. The maps were subsequently smoothed with a 2D Gaussian kernel with s.d. of two bins (4 cm) in both the *x* and the *y* directions. For FR maps and spike plots from a large number of cells recorded in the parametric experiments, see Supplementary Figs. 10–13.

**Spatial information content**. To calculate the spatial information content (Fig. 1c, e) we used the cell's FR map to compute the spatial information rate[38] as

$$\sum_{i=1}^{N} p_i \frac{\lambda_i}{\lambda} \log_2 \left( \frac{\lambda_i}{\lambda} \right)$$

where $\lambda_i$ is the mean FR in the *i*th bin, $\lambda$ is the overall mean FR and $p_i$ is the probability of the mouse being in the *i*th bin (time spent in the *i*th bin/total recording time). Spatial information content in bits per spike was obtained by dividing the information rate by the mean FR of the cell.

**OV score**. To calculate the OV score (Fig. 1d, f) we first calculated FR maps that expressed the cell's firing as a function of the animal's distance (cm) and orientation (degrees) from the object (Fig. 1b) ('vector-maps' in ref. [23]). For this, we sorted spikes and position estimates into distance bins of 2 cm and orientation bins of 10°. The FR in each bin was calculated by dividing the number of spikes by the amount of time spent in the bin. The maps were circularly smoothed with a 2D Gaussian kernel with s.d. of 1.5 bins (3 cm, 15°). In these maps, East (North) relative to the room's frame was defined as 0° (90°). Having computed these FR maps, the OV score was defined as the Pearson correlation between maps from the 'Object' and 'Moved Object' trials (Fig. 1b). When the cell's firing depends on the animal's position relative to the object, rather than the animal's absolute position, the correlation should be high.

**Shuffling of spike data**. In order to identify OV cells with (1) spatial firing patterns and (2) vectorial firing patterns locked to the object, we required thresholds (i.e. cutoff values) for (1) spatial information content and (2) the OV score. To find such thresholds, we implemented a standard shuffling procedure. To shuffle the spikes of the cell, the entire sequence of spike timestamps was shifted in time relative to the mouse's path by a random number. The random number was drawn from a uniform distribution such that

$$p(t_{\text{shift}}) = \frac{1}{b - a}$$

where
  $b$ = session length–20 s
  $a$ = 20 s
  For all $t_{\text{shift}}$ such that 20 s $\leq t_{\text{shift}} \leq$ session length – 20 s
The shifted spike timestamps were wrapped around from the end of the trial, ensuring that all shuffled spikes occurred at some point during the trial. Time shifts varied randomly between permutations and cells.
  We performed cell-by-cell shuffling[33,39] rather than global shuffling[23,40,41]. For each cell, we performed 200 shuffles of the cell's spike timestamps from the 'Object' trial. We then calculated the spatial information content for each one of these shuffles, yielding the cell-specific shuffling distribution (Fig. 1c). The 99th percentile of this shuffling distribution was taken as the cell-specific threshold. That is, cells with actual spatial information content greater than the 99th percentile of the shuffling distribution were considered spatially modulated cells (Fig. 1e). Almost the same procedure was implemented for the OV score, with the only exception that we shuffled spike timestamps from both the 'Object' and 'Moved Object' trials separately, generating pairs of shuffled cells. We then calculated the OV score based on each pair of shuffles.

**Definition of OV cells**. As in earlier work[23], cells had to satisfy multiple criteria to classify as OV cells:

(1) (Pre-selection criterion for new fields) Since objects should induce the presence of new firing fields in OV cells, we required one or more new fields to appear in the 'Object' trial compared to the 'Empty Box' trial. For this, we used the Bayesian field detection algorithm to identify all fields present in the 'Empty Box' and 'Object' trials. We then compared distances between field centres in the two trials. Fields with centres < 2 cm apart were considered the same field. Using this procedure, we verified that each cell had at least one new field in the 'Object' trial.

(2) (Spatial information criterion) Since OV cells should show spatial tuning when objects are present in the environment, we required the spatial information content in the 'Object' trial to exceed a cell-specific threshold. This cell-specific threshold was the 99th percentile of the cell's own shuffling distribution (Fig. 1c). We created this shuffling distribution by shifting the cell's spike timestamps 200 times (see 'Shuffling of spike data') and calculating the spatial information for each one of the shuffled cells.

(3) (OV score criterion) Since the OV score quantifies how strongly the cell's firing depends on the animal's distance and orientation from the object (see 'OV score') we required the OV score to exceed a cell-specific threshold. The threshold was the 99th percentile of the cell's own shuffling distribution (Fig. 1d). We created this distribution by separately shifting the cell's spike timestamps 200 times in the 'Object' and 'Moved Object' trials and calculating the OV score for each shuffled pair.

(4) (Field distance criterion) In order for the cells to be 'vectorial', the cell's firing must be offset from the object rather than located directly at the object. Therefore, as in previous work[23], we required that OV cells had fields

offset from the object's centre by more than 4 cm. To implement this criterion, we detected firing fields on the 'Object' trial and measured the distance of their centres from the object. Only cells that met the 4 cm distance criterion were kept. We note that cells that fire directly at the object are rare in MEC. In this dataset, we observed only a single cell that passed all other criteria for OV cells but failed to pass the distance criterion.

**Field detection algorithm.** Since the aim of the study was to investigate the most basic features of objects that produce activity in OV cells, we expected some primitive stimuli to elicit weak firing fields. Consistent with this, in many OV cells, we found that stimuli consisting of visual contrasts produced firing fields that were barely noticeable (Supplementary Fig. 5a). To detect weak firing fields, in the presence of background noise, we required a sensitive field detection algorithm. For this purpose, we developed a Bayesian field detection algorithm (Supplementary Fig. 3). The overall idea of the algorithm is to evaluate every $x, y$ location in the environment (in $1 \times 1$ cm bins) to determine how likely each location is to contain the firing field of the cell (given the spike data of the cell). We first compute a probability distribution for the fields (step 1), so that we have a probability value for each $x,y$ location in the environment. We then infer the firing fields of the cell by extracting the local peaks of the probability distribution (step 2). Note that the peaks must be sufficiently far apart to classify as two different fields (we used a threshold of 10 cm).

*Step 1. Probability distribution for field locations.* The overall goal can be described by Bayes' theorem. We want to calculate

$$P(\text{field}_{x,y}|D) = \frac{P(D|\text{field}_{x,y})P(\text{field}_{x,y})}{P(D)}$$

where we define

field$_{x,y}$ ≡ the cell's firing field lies at the location $x, y$

$D$ ≡ the cell's spike data

In words, we want to calculate the probability that the cell's field lies at a particular location $(x, y)$ given the spike data of the cell. This is given by the likelihood that the cell's spikes were produced by a field lying at this location (the term $P(D|\text{field}_{x,y})$, multiplied by the prior probability that the field lies at this location (the term $P(\text{field}_{x,y})$), divided by a normalising constant (the term $P(D)$).

Our goal in step 1 is to calculate a probability distribution for the field locations, which requires us to assign a likelihood and choose a prior. For the likelihood, which is a model for the process that generates the data, we assume that spikes originate from two sources: (a) a 2D Gaussian that represents the cell's firing field and (b) a uniform non-zero floor that represents the cell's tendency to produce noisy spikes anywhere in the environment (Supplementary Fig. 3a).

All positions in the animal's environment can be represented in discrete steps (we used 1 cm bins) by

$$\text{pos} = \begin{pmatrix} \text{pos}_1 \\ \text{pos}_2 \\ \vdots \\ \text{pos}_N \end{pmatrix} = \begin{pmatrix} x_1 & y_1 \\ x_2 & y_2 \\ \vdots & \vdots \\ x_N & y_N \end{pmatrix}$$

We want the algorithm to calculate the probability that the spike data originate from a firing field at each of the locations $\text{pos}_1, \text{pos}_2 \ldots \text{pos}_N$. Each iteration of the algorithm corresponds to calculating the likelihood for one such location.

For $\text{pos}_1 = (x_1, y_1)$ (first row of the matrices above), the first iteration of the algorithm proceeds as follows. Let the spike data be represented by

$$D = \begin{pmatrix} d_1 \\ d_2 \\ \vdots \\ d_n \end{pmatrix} = \begin{pmatrix} x_1^{\text{spikes}} & y_1^{\text{spikes}} \\ x_2^{\text{spikes}} & y_2^{\text{spikes}} \\ \vdots & \vdots \\ x_n^{\text{spikes}} & y_n^{\text{spikes}} \end{pmatrix}$$

where the first column represents the $x$-coordinate at which each spike occurred and the second column represents the $y$-coordinate. First, we will apply a selector function that picks out a subset of all spikes, i.e., the function will select spikes that occurred in a circle (of radius $r$) centred around $\text{pos}_1 = (x_1, y_1)$. This step ensures that the results are not influenced by spikes that occurred far away from the current location $\text{pos}_1 = (x_1, y_1)$. More precisely, we use the reduced spike matrix

$$\hat{D} = \begin{pmatrix} \hat{d}_s \\ \vdots \\ \hat{d}_{s+m} \end{pmatrix}$$

where

$\hat{d}_i = d_i$

For all $d_i$ such that

$$\sqrt{(d_i - \text{pos}_1)(d_i - \text{pos}_1)\top} < r$$

where we chose $r = 15$ cm. That is, $\hat{d}_s$ is the first spike that falls within a circle of radius r centred at $\text{pos}_1$ and $\hat{d}_{s+m}$ is the last spike that falls within the circle. $m$ is the total number of spikes that fall within the circle.

Using only the spikes in a circle around $\text{pos}_1$, we calculate the likelihood as

$$\mathcal{L}(\text{pos}_1) = (b + \exp(-(\hat{d}_s - \text{pos}_1)\tfrac{\Sigma^{-1}}{2}(\hat{d}_s - \text{pos}_1)\top))$$
$$\times (b + \exp(-(\hat{d}_{s+1} - \text{pos}_1)\tfrac{\Sigma^{-1}}{2}(\hat{d}_{s+1} - \text{pos}_1)\top)) \cdots$$
$$\times (b + \exp(-(\hat{d}_{s+1} - \text{pos}_1)\tfrac{\Sigma^{-1}}{2}(\hat{d}_{s+1} - \text{pos}_1)\top))$$

where $b = 1.1$ is a constant that implements the uniform non-zero floor (Supplementary Fig. 3a), $\top$ denotes the transpose and $\Sigma^{-1}$ is the inverse of the covariance matrix, which was the identity matrix. We have assumed independence between data points, allowing us to multiply the terms corresponding to the different data points.

In practice, we calculate the log likelihood

$$\log(\mathcal{L}(\text{pos}_1)) = \log(b + \exp(-(\hat{d}_s - \text{pos}_1)\tfrac{\Sigma^{-1}}{2}(\hat{d}_s - \text{pos}_1)\top))$$
$$+ \log(b + \exp(-(\hat{d}_{s+1} - \text{pos}_1)\tfrac{\Sigma^{-1}}{2}(\hat{d}_{s+1} - \text{pos}_1)\top)) \cdots$$
$$+ \log(b + \exp(-(\hat{d}_m - \text{pos}_1)\tfrac{\Sigma^{-1}}{2}(\hat{d}_m - \text{pos}_1)\top))$$

To evaluate the log likelihood, we subtract the maximum value of the log likelihood as follows

$$\exp(\log(\mathcal{L}(\text{pos}_1)) - \max(\log(\mathcal{L}(\text{pos}_1))))$$

which yields the likelihood $\mathcal{L}(\text{pos}_1)$ that the field is at the location $\text{pos}_1 = (x_1, y_1)$. Repeating this process for locations $\text{pos}_2 \ldots \text{pos}_N$ gives the full likelihood distribution $\mathcal{L}(\text{pos}_1) \ldots \mathcal{L}(\text{pos}_N)$. As prior, we chose a Gaussian centred at the object location. That is, for $\text{pos}_1 = (x_1, y_1)$, the prior probability was

$$P(\text{field}_{x_1,y_1}) = \exp(-(\text{pos}_1 - \text{object})\tfrac{\Lambda^{-1}}{2}(\text{pos}_1 - \text{object})\top)$$

where object $= (x_{\text{object}}, y_{\text{object}})$ are the coordinates of the object and $\Lambda = 400 \times \begin{pmatrix} 1 & 0 \\ 0 & 1 \end{pmatrix}$. These values were chosen to match the distribution of OV fields observed in previous work[23]. Specifically, the prior accounts for the experimental finding that OV fields are more common near than far away from the object. This property has also been observed for BV cells[14,15,42]. Finally, we multiply the likelihood by the prior and normalise to find the posterior probability distribution.

*Step 2. Extracting field positions from probability distribution.* Once we had acquired a probability distribution for the cell's fields, the aim was to extract field positions from the probability distribution. This step can be implemented in multiple ways, but we used a simple approach based on finding local maxima in the probability distribution. As a preliminary note, suppose that OV cells were known only to have a single firing field. In that case, we would have simply found the maximum of the probability distribution and used the associated x, y coordinates as the field location. That is,

$$\hat{\text{field}}_{x,y} = \underset{\text{field}_{x,y}}{\arg\max}(P(\text{field}_{x,y}|D))$$

where $\hat{\text{field}}_{x,y}$ is the estimated field location. However, we included the possibility that OV cells had two or more firing fields, as known from previous work[23]. For this, we extracted the 50 probability values that maximised the posterior and sorted them in descending order, as follows:

$$p_1 \geq p_2 \geq \cdots \geq p_{50}$$

where $p_1$ is the largest value in the probability distribution, $p_2$ is the second largest value and so on. We had 50 position coordinates associated with these probabilities, which we denote by $c$

$$c_1, c_2, \cdots c_{50}$$

Some of these coordinates will be neighbours and likely represent the same field. To remove close neighbours, we calculated a Euclidian distance matrix $\mathcal{M}$ where each element $\mathcal{M}_{ij}$ is the Euclidian distance function (denoted by $d$) between coordinates $c_i$ and $c_j$

$$\mathcal{M}_{ij} = d(c_i, c_j) = \sqrt{(c_i - c_j)(c_i - c_j)\top}$$

Because the Euclidian distance function is symmetric, we considered only the upper triangular part of this matrix. We considered two coordinates $c_i$ and $c_j$ as neighbours if

$$d(c_i, c_j) < 10 \text{ cm}$$

For each probability value, we therefore had a set of neighbours. If a probability value was larger than all its neighbours, that probability value (and the associated $x$, $y$ coordinates) was taken as an estimated field position. This process was repeated for all probability values until all field positions had been extracted.

*Simulations.* In order to verify the algorithm on data where we had ground truth (knew the real field location), we simulated cells with spatial firing fields and noisy

background activity (Supplementary Fig. 3b, c). To simulate the firing field, spikes were drawn randomly from a 2D Gaussian centred at some random location in a $100 \times 100$ cm$^2$ environment. The number of spikes from the field was 500, 250, 125, 50 or 15, depending on the condition. To simulate background noise, we drew spikes from a uniform distribution across the $100 \times 100$ cm$^2$ environment. The number of spikes drawn from the noise was 0, 2500, 25000, 50000 or 100000, depending on the condition. Visual inspection confirmed that the algorithm successfully found firing fields even in cases where they were completely obscured by noise (Supplementary Fig. 3bi–iii). To systematically test algorithm performance, we performed 50 repetitions for each condition. To quantify the error of the algorithm, we calculated the Euclidian distance between the algorithm's estimate of the field location and the true field location

$$\text{error} = \sqrt{(\text{estimate}_{x,y} - \text{true}_{x,y})(\text{estimate}_{x,y} - \text{true}_{x,y})\top}$$

The algorithm's estimate was taken as the maximum of the posterior probability distribution

$$\text{estimate}_{x,y} = \underset{\text{field}_{x,y}}{\text{argmax}}(P(\text{field}_{x,y}|D))$$

Performance was perfect (mean error 0 cm) across all levels of noise when the field contained 500 or 250 spikes (Supplementary Fig. 3c). When the field contained 125 spikes, the mean error was lower than 2 cm for all levels of noise except the largest one (mean error 11.8 cm). Large errors were only found for conditions in which the firing field consisted of a small number of spikes (50 or 15 spikes). The chance level (40.21 cm) was taken as the expected error of an algorithm making random (uniformly distributed) estimates of the field location.

### Measures of responsiveness of OV cells

*Firing rate inside a circular ROI.* In order to assess how strongly OV cells produce vectorial responses to different stimuli, we used two measures of responsiveness: (1) the FR of the cell inside a circular ROI in which we expected the cell to fire based on its vector coordinates and (2) the OV score. For measure (1), we first applied the Bayesian field detection algorithm (Supplementary Fig. 3) to the 'Object' trial to identify the coordinates of the cell's firing fields. By subtracting the object location, we obtained object-referenced coordinates of the firing fields, which we term 'vector coordinates'. The vector coordinates were transferred to the experimental trial (one of the trials in the 2D/3D, transparent or contrast experiment). The ROI was taken as a circle of radius 15 cm, centred at the vector coordinates. For how this parameter choice (size of the ROI) affects the experimental findings, see Supplementary Fig. 4. All position samples and spikes that fell within the ROI were extracted. The number of spikes inside the ROI was divided by the amount of time the animal spent inside the ROI, to obtain the FR.

*The OV score.* For measure (2), we had to calculate FR maps that expressed the cell's firing as a function of the animal's distance and orientation to the object (Fig. 1b) (object-centred FR maps: see 'OV score' for details). The maps were calculated in the 'Object' trial and the experimental trial (one of the trials in the 2D/3D, transparent-object or contrast experiment). We then calculated the OV score for this pair of trials, as the Pearson correlation between the object-centred maps. If the cell's firing in the experimental trial correlated with the firing in the 'Object' trial—that is, if the distance and direction tuning was preserved—the OV score was expected to be high. We note that for 2D stimuli on the wall of the arena the OV score suffers from a binning problem: the animal can only explore half of all orientation bins around the object (180° out of the full 360°). For this reason, we give stronger weight to measure (1) than measure (2) in the present study.

### Inferring the probability that OV cells respond to different objects.

In order to compute (1) the probability that OV cells respond to different objects and (2) a range of reasonable FR values for each object, we performed a Bayesian statistical analysis (Fig. 5). The method is standard[26,27] and based on similar principles as the field detection algorithm (Supplementary Fig. 3). Intuitively, suppose we have a dataset ($n = 4$) that consists of FRs of OV cells for some object, and suppose that the data points cluster around 3 Hz. For example, the data points might be 2.7, 2.9, 3.1 and 3.3 Hz. This dataset would be likely to have been produced by a Gaussian at 3 Hz but unlikely to have been produced by a Gaussian at 20 Hz. In other words, the Gaussian with mean at 3 Hz is a better explanation of the dataset than the Gaussian with mean at 20 Hz. The idea behind the method is to generalise this process and systematically move the Gaussian across a range of plausible mean values (for example, from −10 to 10 Hz in steps of 0.01), to see which values are most consistent with the actual data obtained. This assumes a Gaussian as the underlying model for the process of data generation, but we verified that our results were insensitive to this assumption (see below). For ease of illustration, suppose we perform the analysis for the white contrast condition. To compute (1) and (2), we would like to calculate $P(FR|D)$—the probability that OV cells respond with a particular change in FR to the white contrast, given the data ($D$) obtained (Figs. 2c, 3c and 4c). According to Bayes' theorem, we can write this as

$$P(\text{FR}|D) = \frac{P(D|\text{FR})P(\text{FR})}{P(D)}$$

where $P(D|FR)$ is the likelihood of a particular FR and $P(FR)$ is the prior probability of that FR. More precisely, we defined the data ($D$) as all the FR values observed for OV cells inside their ROI (relative to the FR in the same ROI in the 'Empty Box' trial)

$$D = \begin{pmatrix} d_1 \\ d_2 \\ \vdots \\ d_n \end{pmatrix}$$

That is, $d_1$ is the FR of the first OV cell inside its ROI in the 'White contrast' trial (subtracted by the same value in the 'Empty Box' trial), $d_2$ is the FR of the second OV cell inside its ROI in the 'White contrast' trial (subtracted by the same value in the 'Empty Box' trial). $n$ is the total number of OV cells. For the results presented in Fig. 5, we assigned a Gaussian likelihood. The likelihood was calculated as

$$p(D|\text{FR}) = \frac{1}{(\sqrt{2\pi}\sigma)^n} \exp\left(-\frac{1}{2}\sum_i^n \frac{(d_i - \text{FR})^2}{\sigma^2})\right)$$

where FR is the mean of the Gaussian and $\sigma$ is the standard deviation. FR is the parameter we try to infer—the estimated change in the FR for the white contrast (compared with the 'Empty Box' session). The key question of interest is, 'What are the most probable values of this parameter, given the data?'. To choose a value for $\sigma$, we used the standard deviation observed in the real data. Given that most of our OV cells were recorded on different days and in different animals, we assumed independence between data points. This allows for summation over different $d_i$ inside the exponential. The calculation above was repeated for values of FR from −10 to 10 Hz with a step size of 0.01 Hz. That is, we repeated the calculation for all reasonable values of FR. Note that we did not observe FR increases of more than 10 Hz in the real data. We assumed a uniform prior, given by

$$p(\text{FR}) = \frac{1}{b - a}$$

where $b = 10$ was the upper bound of FR and $a = -10$ was the lower bound. That is, all FR values had the same prior probability before seeing the data. Multiplication of the likelihood by the prior, followed by normalisation, gave the probability distribution $P(FR|D)$. In order to calculate (1) the probability that OV cells responded to the object, we calculated the sum of all probabilities to the right of the origin

$$p(\text{response}) = \sum_{\text{FR}>0} P(\text{FR}|D)$$

In order to calculate (2) a range of reasonable FR values for each object, we sorted all probabilities in descending order and summed them until the amount of probability exceeded 0.95. Looking at the FR values associated with these probabilities, we used the minimum FR and the maximum FR as the lower and upper bound, respectively. This gives a 95% probability range for the true FR ('credible region'[27]).

In Supplementary Fig. 7, we tested the sensitivity of the results of this analysis to different priors. First, we used a Gaussian prior (Supplementary Fig. 7a). The prior probability was calculated as

$$p(\text{FR}) = \frac{1}{\sqrt{2\pi}\sigma} \exp\left(-\frac{(\text{FR} - \mu)^2}{2\sigma^2}\right)$$

where we set $\mu = 0$ an $\sigma = 2$.

We then used priors based on the beta distribution (Supplementary Fig. 7b, c). Because the beta distribution is defined for $x$ on the interval [0,1] we first mapped each value of FR to this interval. That is, the smallest value FR = −10 Hz was mapped to 0, the largest value FR = 10 was mapped to 1, and so on. The prior probability was calculated as

$$p(\text{FR}) = p(x) = \frac{\Gamma(\alpha)\Gamma(\beta)x^{\alpha-1}(1-x)^{\beta-1}}{\Gamma(\alpha+\beta)}$$

where $\Gamma$ is the gamma function. In panel b, we used $\alpha = 3$ and $\beta = 1.5$. In panel c, we used $\alpha = 1.5$ and $\beta = 1.5$.

In Supplementary Fig. 8, we tested the sensitivity of the results of the analysis to different likelihoods. We first used a binomial likelihood (Supplementary Fig. 8a). To compute the binomial likelihood, we first binarized the data by applying the sign function

$$\text{sign}(d_i) = \begin{cases} 1 \text{ if } d_i > 0 \\ -1 \text{ if } d_i < 0 \end{cases}$$

This gave $r$ successes (number of OV cells with increased FR) and $(n - r)$ failures (number of OV cells with decreased FR) in $n$ trials (total number of OV cells recorded). The aim was to infer the 'bias' of OV cells to respond to the object. This is analogous to inferring the bias of a coin after observing $r$ heads and $(n - r)$ tails in $n$ trials. The likelihood was calculated as

$$p(D|y) = \frac{n!}{r!(n-r)!}y^r(1-y)^{n-r}$$

where $y$ is the bias, $y \in [0, 1]$. Next, we used a Cauchy likelihood (Supplementary Fig. 8b). The Cauchy distribution is a unimodal distribution like the Gaussian but has much fatter tails. We took this as a reasonable test of the sensitivity of the results to the shape of the likelihood function. In this case, we first calculated the likelihood given two parameters (the median 'FR' and the half-width '$b$')

$$p(D|\text{FR}, b) = \frac{1}{\pi}\frac{b}{(b^2 + (d_1 - \text{FR})^2)} \times \frac{1}{\pi}\frac{b}{(b^2 + (d_2 - \text{FR})^2)} \times \cdots \frac{1}{\pi}\frac{b}{(b^2 + (d_n - \text{FR})^2)}$$

We then calculated the marginalised likelihood

$$p(D|\text{FR}) = \sum_b p(D|\text{FR}, b)$$

The pattern of results from the Bayesian statistical analysis (Fig. 5) was similar (or stronger) when changing the prior (Supplementary Fig. 7) and when changing the likelihood function (Supplementary Fig. 8) confirming that the results in the main analysis were insensitive to the precise assumptions made (a Gaussian likelihood and a uniform prior; Fig. 5).

**Histology and reconstruction of recording positions**. The tetrodes were not moved after the final recording session. The mouse was given an overdose of pentobarbital and was perfused intracardially with 9% saline and 4% formaldehyde. The brain was extracted and stored in 4% formaldehyde. Frozen, 30-mm sagittal sections were cut, mounted on glass, and stained with cresyl violet (Nissl). The final position of the tip of each tetrode was identified on photomicrographs obtained with an Axio Scan.Z1 microscope and Axio Vision software (Carl Zeiss) (Supplementary Fig. 1).

**Statistical tests**. All statistical tests were two-sided. We used Kruskal–Wallis tests for variance analysis between groups and Wilcoxon signed-rank tests for paired tests. Correlations were determined using Pearson's product–moment correlation coefficients. No statistical methods were used to pre-determine sample sizes but our sample sizes are similar to those reported in previous publications. The study contained no randomisation to experimental treatments and no blinding.

**Reporting summary**. Further information on research design is available in the Nature Research Reporting Summary linked to this article.

## Data availability

Spike- and position data are publicly available at the following https://doi.org/10.6084/m9.figshare.16560126.v1. Source data underlying all main figures are provided in Supplementary Data 1. Other data are available upon request from the corresponding authors.

## Code availability

Custom code used in this paper is available upon request from the corresponding authors.

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

## Acknowledgements

We thank Ø. Høydal for discussion and sharing code, J. Carpenter and B. Kanter for proofreading and L. Porta Mana for discussion of the field detection algorithm. We thank A.M. Amundsgård, N. Dagslott, K. Haugen, K. Jenssen, E. Kråkvik, I. Ulsaker-Janke and H. Waade for technical assistance. The work was supported by an RCN FRIPRO grant to M.-B.M. (grant number 300394), a Centre of Excellence scheme grant to M.-B.M. and E.I.M. and a National Infrastructure grant to E.I.M. and M.-B.M. from the Research Council of Norway (Centre of Neural Computation, grant number 223262; NORBRAIN, grant number 295721), the Kavli Foundation (M.-B.M. and E.I.M.), a grant from Stiftelsen Kristian Gerhard Jebsen (K.G. Jebsen Centre for Alzheimer's Disease; grant number SKGJ-MED-022) and a direct contribution to M.-B.M. and E.I.M. from the Ministry of Education and Research of Norway.

## Author contributions

S.O.A., M.-B.M. and E.I.M. designed experiments and interpreted data; S.O.A. performed experiments and performed all analyses; S.O.A. wrote the paper, with assistance from E.I.M. and M.-B.M.; E.I.M. and M.-B.M. supervised the project.

## Competing interests

The authors declare no competing interests.
