## [Transparent Peer Review File · Communications Biology]

Reviewers' comments:

Reviewer #1 (Remarks to the Author):

This paper reports an analysis of neurons recorded from the medial entorhinal cortex of freely exploring mice, investigating a class of recently discovered cells that respond most strongly at a given distance and direction from objects in the environment - so-called object vector cells. In the present study, the stimuli eliciting object vector responses are investigated in three experiments. In the first, the objects were varied between fully 3D, partly 3D and fully 2D. In the second, the object was replaced by a transparent window in the wall, of the same size as the object, and in the third the window was replaced with a card of the same size that was dark gray, pale gray or white (the walls were black). It was found that object vector responses persisted with these manipulations but varied in intensity as the stimuli became progressively less salient. It is concluded that object vector cells are anchored by visual features, rather than 3D objects per se.

This is a nicely written paper and these are well-motivated experiments representing a natural step towards understanding what it is that these neurons are sensitive to, and the result is interesting and slightly surprising. I don't have major concerns about the analyses except that the data set is very small - only 30 neurons from 5 animals, which means that these are preliminary results rather than a completed finding. They are, however, provocative and will stimulate further research. We need to know how many neurons came from each animal (to be sure that these observations are general), what percentage of total neurons these represented, and in what experiment each neuron was recorded in.

Given the small number of neurons, it would be good to see the individual data sets, perhaps in the Supplementary materials. I prefer spike plots like the ones in Supp. Fig. 3 as one gets a better sense of the actual data. I appreciate the efforts to quantitatively isolate the "signal" from the "noise" in these data, but these could be explained more intuitively for the non-mathematical reader as the many equations are rather formidable.

Other comments:

Were the stimulus types presented in a systematic or randomized order? If the former, might there have been temporal order effects? Did the mice respond behaviorally to the changes, as when the object's projection from the wall was varied?

L25 How many cells did vs. did not respond to visual contrast?

The last paragraph of the discussion was speculative and off-topic and I did not feel added to the picture.

L359 The description of vertical stacking was confusing - perhaps better to just say "circularly smoothed". I wonder why these data, being a distance and direction, aren't presented on a polar plot, which would seem to be more intuitive?

Fig. 2 - a minor point, but the perspective of the drawings is wrong and the walls look like they are slanted backwards. I think it would be more consistent to also use the same viewpoint for the stimuli in Fig. 4, and to rotate the firing rate maps so that the objects are at the top, as they are in the previous figures.

Reviewer #3 (Remarks to the Author):

This paper builds on the prior discovery of object vector cells (Hoydal et al, 2019) and sets out to characterize fundamental features these 'OVs' are responding to. The authors used single unit recording in medial entorhinal cortex to assess the activity of OV cells as mice were freely exploring an arena containing items ranging from free-standing 3-dimensional objects to two-dimensional surfaces. They compared the responses of OV cells as the characteristics of the item varied. They propose that visual contrast is the minimal stimulus that elicits activity in these OV cells, with a higher probability of response for high contrast – essentially that 3D structure is not necessary to drive OV activity.

This study refines our understanding of OV cells which are of considerable and timely interest. Knowing that OV cells respond to visual contrast and may represent a significant proportion of the previously described aperiodic cells is of valuable interest for the spatial navigation community. The study is thoroughly conducted and overall convincing. Methods are detailed, systematic, and clear – especially so the description of the approaches used for field detection and the Bayesian statistical analysis of the OV cell responses. I have identified a small number of points – which I consider to be minor. I would like to see these issues addressed before publication but beyond that would be happy for this interesting work to be published.

Minor points

1. Parametric experiments: It is not clear enough that OV score is computed relative to the initial 'Object Trial', with a fully exposed 3D object (and not to a moved 2D object for example). In order to clarify the way OV score is computed, it would be helpful to

- show the rate maps of the corresponding 'Object Trial' map in the examples (Figures 2b ,3b, 4b)
- make this point clearer in the text (L80)

2. The authors propose that visual contrast is the fundamental feature of an 'object' driving OV cell activity. Although they convincingly show that visual contrast elicits OV activity, the contribution of tactile cues cannot be completely excluded. Indeed, Hoydal et al 2019 showed that in the absence of visual cues OV cell tuning degrades but does not disappear (FigS7d-e), which could potentially be due to the presence of tactile information in the dark. In the present study, tactile cues have been minimized, but the detection of different textures (wall vs transparent film or vs adhesive tape) by the mice cannot be excluded. Thus, other fundamental features of objects might also be able to elicit OV cells firing. I do not require the authors to conduct further analyses but they should mention this potential confound in their discussion.

3. Responsiveness to low contrast:

The authors claim that "a visual contrast is sufficient for OV cells to respond, with increasing certainty for higher contrasts." The authors should discuss if the variability in OV responses to low contrast stimuli is a direct encoding of the perceptual quality of the object or if it might simply reflect a failure of the mice to detect the visual contrast at some time points. One resolution would be if the authors had simultaneously recorded cells responding and cells not responding to a low contrast object (this would exclude the second scenario).

4. L246-250 "That is, spatial representation in MEC might be functionally separated into two distinct systems: one involving grid cells, speed cells and HD cells (dependent on PV interneurons via grid cells and speed cells³⁵) for self-motion-dependent representation in open spaces^{2,36} and one involving OV cells and border cells (putatively dependent on SOM interneurons via OV cells) for more landmark-based navigation. "

Whereas such a dichotomy is appealing, I cannot see in Miao et al. 2017 elements allowing to support the claim 'dependent on PV interneurons via grid cells and speed cells³⁵' and 'putatively dependent on SOM interneurons via OV cells'. Instead, this study shows that PV or SOM interneuron inactivation has no effect on head-direction cells and border cells. Could the authors make this statement more nuanced or add complementary references?

5. Typos:

Main text

a. In the following sentences, reference 8 does not seem the correct one (Kropff et al. 2015, Speed cells in the MEC):

L41 "In the latter pair of trials, the object was a 40×8×8 cm (height × base area) multi-colour Duplo tower, known to produce clear OV responses⁸ (Supplementary Fig. 2)"

L46. As in earlier work⁸, cells had to pass multiple criteria to count as OV cells.

b. L52. "the firing fields were displaced > 4 cm away from the object centre."
'Displaced' is slightly confusing, 'positioned' would be more appropriate.

c. L126. "While some cells had firing rates that clearly increased with visual contrast (Fig. 4b) other cells were more difficult to evaluate by eye (Supplementary Fig. 5a)"

Specify which firing rate (Is it inside ROI? overall FR?).

Referring to evaluation by eye is slightly confusing.

Methods section

L259: buprenorphine, meloxicam: specify why these drugs are injected.

L261: buprenorphine, meloxicam, bupivacaine: specify doses.

Legends

Fig S8:

- Rationalize the way the number of OV cells are denoted (n in the methods and N in the legend of Fig S8)

- The authors probably wanted to say 'a firing rate change < 0 Hz' in the following sentence: "N is the total number of OV cells we have data from, r is the number of OV cells that respond with a firing rate > 0 Hz and N - r is the number of OV cells that respond with a firing rate < 0 Hz."

Reviewers' comments:

Reviewer #1 (Remarks to the Author):

This paper reports an analysis of neurons recorded from the medial entorhinal cortex of freely exploring mice, investigating a class of recently discovered cells that respond most strongly at a given distance and direction from objects in the environment - so-called object vector cells. In the present study, the stimuli eliciting object vector responses are investigated in three experiments. In the first, the objects were varied between fully 3D, partly 3D and fully 2D. In the second, the object was replaced by a transparent window in the wall, of the same size as the object, and in the third the window was replaced with a card of the same size that was dark gray, pale gray or white (the walls were black). It was found that object vector responses persisted with these manipulations but varied in intensity as the stimuli became progressively less salient. It is concluded that object vector cells are anchored by visual features, rather than 3D objects per se.

This is a nicely written paper and these are well-motivated experiments representing a natural step towards understanding what it is that these neurons are sensitive to, and the result is interesting and slightly surprising. I don't have major concerns about the analyses

We are grateful to receive such positive feedback from the Reviewer. In addition, we thank him or her for the insightful comments. We have modified the article in response to essentially all of them, leading to many improvements in the article.

except that the data set is very small – only 30 neurons from 5 animals, which means that these are preliminary results rather than a completed finding. They are, however, provocative and will stimulate further research.

We understand that one might get the impression that the dataset is relatively small. However, in reality the data are richer and more extensive than suggested by the above cell number alone:

- (1) First, the dataset consisted of 67 OV cells from 7 animals, distributed across the experiments (not 30 cells from 5 animals, as suggested by the Reviewer). The confusion was probably caused by us not clearly specifying where to look for this information in the previous version of the manuscript. In the revised version, this information should be much easier to find (see our reply to next comment).
- (2) It is important to remember that 67 cells refers to the number of OV cells. In the present study, OV cells correspond to 13.6% of the recorded cells, so that the total number of cells is 492. We thank the Reviewer for pointing out that this information was lacking. In the revised manuscript, we include the total cell number in the first paragraph of the Results.
- (3) The number of trials performed on each cell is extensive. First, we identify OV cells in 3 trials ('Empty Box', 'Object', 'Moved Object'). Second, we conduct parametric

experiments consisting of a total of 11 trials ($4 + 4 + 3 = 11$, for the 2D/3D-, transparent- and contrast experiment, respectively). The within-subject nature of the design compensates for the apparently low number of cells (which we agree would have been low if the study had a standard between-subject design).

- (4) Statistical analysis. Suppose we did not have enough cells and had acquired too little data. We believe that both statistical approaches we used would have picked this up. In the traditional statistical approach, we would have been implicitly told that the data are not enough by finding non-significant effects (in contrast, we found significant main effects for all experiments). In the Bayesian statistical analysis, we would have been visibly told to collect more data by finding broad curves failing to tell us what the true firing rate might be. In contrast, the probability distributions were peaked at a narrow range of firing rate values (Fig. 5). Our results were also insensitive to the choice of prior (Supplementary Fig. 7), a typical sign that the amount of data is sufficient.
- (5) Sample sizes compared to earlier work. The number of cells in our experiments is similar to that of earlier studies such as Høydal et al. (2019) in Nature.

We need to know how many neurons came from each animal (to be sure that these observations are general), what percentage of total neurons these represented, and in what experiment each neuron was recorded in.

We thank the Reviewer very much for this suggestion, which we agree will help the reader and prevent misunderstandings on how many cells were recorded (see above).

Supplementary Table 1 now shows the total number of cells, the number of OV cells and percentages of OV cells for each animal. It also shows how many cells each animal contributed to each experiment. We refer to this table in the Results, and in the same paragraph we also provide the following additional information about cell numbers:

The mean number of OV cells per mouse was 9.6 (minimum: 2 OV cells; maximum 14 OV cells) (Supplementary Table 1). The lowest fraction of OV cells across all mice was 5.7% while the highest fraction was 16.7%. The fraction of OV cells obtained overall (13.6%) was very similar to the fraction obtained in the previous study recording OV cells in the same region (14.7%)²³.

Given the small number of neurons, it would be good to see the individual data sets, perhaps in the Supplementary materials. I prefer spike plots like the ones in Supp. Fig. 3 as one gets a better sense of the actual data.

In response to this suggestion, we have added two types of plots for the individual cells.

Firstly, to stay consistent with the rest of the paper, we plot rate maps as in Figure 2b, 3b and 4b. Our experience is that converting spikes to firing rates helps visualisation, since for long recording sessions spike plots become very dense and difficult to interpret. In addition, rate maps are normalised with respect to how long time the animal spent in a particular region. This normalisation is not present in spike plots and makes visualisation more difficult. We show rate maps for the individual cells in Supplementary Figure 10 and 11.

Secondly, to comply with the Reviewer's request, we have also added spike plots for the same cells in Supplementary Figure 12 and 13.

I appreciate the efforts to quantitatively isolate the "signal" from the "noise" in these data, but these could be explained more intuitively for the non-mathematical reader as the many equations are rather formidable.

We are grateful that the Reviewer appreciates this, since we did put much effort into developing algorithms for isolating "signal" from "noise". We assume that the Reviewer mainly refers to the Bayesian field detection, which we agree lacked a sufficiently clear and intuitive description in words. We have now added the following text (before the mathematical description):

"The overall idea of the algorithm is to evaluate every x,y location in the environment (in 1×1 cm bins), to determine how likely each location is to contain the firing field of the cell (given the spike data of the cell). We first compute a probability distribution for the fields (step 1), so that we have a probability value for each x,y location in the environment. We then infer the firing fields of the cell by extracting the local peaks of the probability distribution (step 2). Note that the peaks must be sufficiently far apart to classify as two different fields (we used a threshold of 10 cm)."

We hope that this will help readers with less background in mathematics understand the gist of the field detection method (even without reading the mathematical steps in the sections that follow).

We have also expanded the intuitive description of the Bayesian analysis responsible for producing Figure 5. It now reads:

"Intuitively, suppose we have a dataset ($n = 4$) that consists of firing rates of OV cells for some object, and suppose that the datapoints cluster around 3 Hz. For example, the datapoints might be 2.7 Hz, 2.9 Hz, 3.1 Hz and 3.3 Hz. This dataset would be likely to have been produced by a Gaussian at 3 Hz but unlikely to have been produced by a Gaussian at 20 Hz. In other words, the Gaussian with mean at 3 Hz is a better explanation of the dataset than the Gaussian with mean at 20 Hz. The idea behind the method is to generalize this process and systematically move the Gaussian across a range of plausible mean values (for example, from -10 Hz to 10 Hz in steps of 0.01), to see which values are most consistent with the actual data obtained."

Other comments:

Were the stimulus types presented in a systematic or randomized order? If the former, might there have been temporal order effects?

We presented the stimuli in a systematic fashion, in the same order as they appear in the schematics in the figures (for example in Figure 4, starting with no contrast, followed by dark grey, then grey, then white). However, in the first experiment, we performed 27 experiments in the 'standard direction' (2D, then partly embedded, then 3D) and 14

experiments in the 'reverse direction' (3D, then partly embedded, then 2D). (Note that the reason why the number of experiments is larger than the number of cells is because some experiments contained cells that didn't end up passing all the OV cell criteria.) In this experiment, we observed no significant effect of the order of presenting the stimulus types (see Results). The fact that reversing the order did not change the findings rules out temporal order as a major explanatory factor in the 2D/3D experiment.

Did the mice respond behaviorally to the changes, as when the object's projection form the wall was varied?

Our original impression – both from current and past work – was that a) mice are usually very interested in the object and spend a lot of time in the area surrounding it, even if the object is familiar; and (b) the mice's behaviour was similar for different objects. However, we had not quantitatively compared the mice's behaviour for different objects. Since the Reviewer raises the point, we have added Supplementary Figure 9, which provides information about behavioural responses for each of the objects used in the study. We include three behavioural measures:

- (1) The fraction of time the animal spent near the object (25 cm or less away from the object). This is similar for different object types. We note that animals tend to spend less time around visual contrasts compared to other objects, but an equal amount of time around different visual contrasts (i.e., the time spent around the dark grey contrast is roughly the same as the time spent around the grey and white contrasts).
- (2) The mean distance between the animal and object during each trial. Again, the distributions of mean distances were similar for different object types (Supplementary Fig. 9b).
- (3) The percentage of the environment that the animal explored. Specifically, we calculated the percentage of 2x2 cm spatial bins that the animal covered during the trial. The coverage was excellent for all objects (Supplementary Fig. 9c), confirming that differences in the mice's behaviour must have been relatively small and did not prevent us from acquiring good position data from the animals.

Supplementary Figure 9 together with its legend should be self-explanatory but we also added a subsection in the Method under "Behavioural procedures" called "Quantification of behaviour" where we describe these calculations and results.

L25 How many cells did vs. did not respond to visual contrast?

We agree with the Reviewer that this should be included and have put down the numbers in the Results (line 133 and onwards). Pooling across all visual contrasts, OV cells produced an increase in the firing rate inside the ROI in 34 instances. In 8 instances, the cells failed to respond (zero or negative change in firing rate inside the ROI). Note that the sign of the change (response vs. no response) is not the only information that affects the statistical

analysis: it also considers the magnitude of the change. For example, most cells produced a stronger change in firing rate inside the ROI for the white contrast than the grey contrast (and in turn, most cells produced a stronger change for the grey contrast than the dark grey contrast).

The last paragraph of the discussion was speculative and off-topic and I did not feel added to the picture.

We agree that the final paragraph of the Discussion was speculative: it considered implications of the possibility that ‘aperiodic spatial cells’ might be OV cells. The present data raise the possibility that ‘aperiodic spatial cells’ might be OV cells responding to visual cues in an “empty” environment where there are no other stimuli to anchor onto (see the relevant part of the Discussion and Supplementary Fig. 5b and c). One implication of this is that OV cells might be modulated by SOM interneurons. This follows from the finding that ‘aperiodic spatial cells’ are controlled by SOM interneurons (Miao et al. 2016). This would also imply distinct interneuron control of OV cells and grid cells (since grid cells are controlled by PV interneurons).

In the previous version of the manuscript, this paragraph was unnecessarily complicated since we suggested a possible distinction between grid cells/head direction cells/speed cells vs. object-vector cells/border cells. We understand that this felt irrelevant and off topic. It also did not follow directly from the present study. We have now removed references to other cell types than OV cells and grid cells. In addition, we have added a sentence that makes the link to the previous paragraph more explicit:

“The proposal that aperiodic spatial cells could be OV cells has one more implication.”

We hope that – after these changes – readers now will take away from the final paragraph that (a) its main purpose is to suggest the possibility of distinct interneuron control of OV cells and grid cells; and that (b) the suggestion follows directly from the present study since our results raise the possibility that ‘aperiodic spatial cells’ might be OV cells. These suggestions are important pointers for future studies and we would prefer to keep the cleaned-up version of the final paragraph.

L359 The description of vertical stacking was confusing – perhaps better to just say “circularly smoothed”. I wonder why these data, being a distance and direction, aren’t presented on a polar plot, which would seem to be more intuitive?

We have updated the paper in line with the Reviewer’s suggestion. The description of vertical stacking is now removed – this level of detail about the programming was unnecessary (interested researchers will find out themselves how to do circular smoothing).

In the earlier version of the paper, we presented the data in a plot with orientation on the y-axis and distance on the x-axis to make plotting consistent with conventions in Høydal et al.

2019. However, the Reviewer is right that a polar plot would be much more intuitive (for example, in the earlier version, readers would themselves have to imagine that 360 degrees wraps around to 0 degrees). In the revised version of the paper the data are presented in polar plots (Fig. 1b), as the Reviewer suggests.

Fig. 2 – a minor point, but the perspective of the drawings is wrong and the walls look like they are slanted backwards. I think it would be more consistent to also use the same viewpoint for the stimuli in Fig. 4, and to rotate the firing rate maps so that the objects are at the top, as they are in the previous figures.

We have redrawn the schematics in Figure 2, Figure 3 and Figure 4 so that the perspective should hopefully be more accurate, and the walls are no longer slanting backwards. The viewpoint in the schematic in Figure 4 is now the same as in the others.

The reason why the objects are at the bottom in Figure 4 is that this was their real location during the experiment (south). In some other experiment, the location might have been a different one (west, east or north). We always judged (based on the initial screening trials: 'Empty Box', 'Object' and 'Moved Object') where the cell fired relative to the object and then placed the testing-object in an appropriate location during the parametric experiment. For example, if the cell fired to the east of the object, we placed the testing-object (e.g. a 2D object or a visual contrast) on the west wall. This ensured that we didn't lose data by placing the testing-object in a location where it would be impossible for the cell to respond.

We hope this clarifies why the objects are at the bottom in Figure 4 (instead of at the top as in Figure 2 and 3). While it would be fine to rotate the firing rate maps, we think it makes sense to show the real location of the object, and therefore decided to keep the firing rate maps as they were. Hopefully, all of this should make more sense now that we also include the firing rate map from the initial 'Object' trial (according to the suggestion from Reviewer #3).

Reviewer #3 (Remarks to the Author):

This paper builds on the prior discovery of object vector cells (Hoydal et al, 2019) and sets out to characterize fundamental features these 'OVs' are responding to. The authors used single unit recording in medial entorhinal cortex to assess the activity of OV cells as mice were freely exploring an arena containing items ranging from free-standing 3-dimensional objects to two-dimensional surfaces. They compared the responses of OV cells as the characteristics of the item varied. They propose that visual contrast is the minimal stimulus that elicits activity in these OV cells, with a higher probability of response for high contrast – essentially that 3D structure is not necessary to drive OV activity.

This study refines our understanding of OV cells which are of considerable and timely

interest. Knowing that OV cells respond to visual contrast and may represent a significant proportion of the previously described aperiodic cells is of valuable interest for the spatial navigation community. The study is thoroughly conducted and overall convincing. Methods are detailed, systematic, and clear – especially so the description of the approaches used for field detection and the Bayesian statistical analysis of the OV cell responses. I have identified a small number of points – which I consider to be minor. I would like to see these issues addressed before publication but beyond that would be happy for this interesting work to be published.

We thank the Reviewer very much for his or her enthusiasm and for a very detailed reading of the paper, as well as the constructive suggestions.

Minor points

- 1. Parametric experiments: It is not clear enough that OV score is computed relative to the initial 'Object Trial', with a fully exposed 3D object (and not to a moved 2D object for example). In order to clarify the way OV score is computed, it would be helpful to*
 - show the rate maps of the corresponding 'Object Trial' map in the examples (Figures 2b, 3b, 4b)*
 - make this point clearer in the text (L80)*

We agree with both these suggestions, which will help readers to better understand how the OV score is computed.

Firstly, we have tried to describe more clearly that the OV score is computed using the initial 'Object' trial:

"For measure (2), we used the initial 'Object' trial, where the standard Duplo tower was in the middle of the box, as a template (Fig. 1a, middle; Fig. 2a, left: 'Reference trial'). That is, measure (2) was the Pearson correlation between the object-centered rate map from the initial 'Object' trial (Fig. 1b, middle) and the object-centered rate map from one of the experimental trials."

Secondly, we have added the rate maps from the corresponding 'Object' trial in Figure 2b, 3b and 4b, as the Reviewer suggests. We tried to emphasise that this was not part of one of the experimental trials by labelling it 'Reference trial' and by inserting a stippled line to separate it from the experimental trials.

- 2. The authors propose that visual contrast is the fundamental feature of an 'object' driving OV cell activity. Although they convincingly show that visual contrast elicits OV activity, the contribution of tactile cues cannot be completely excluded. Indeed, Hoydal et al 2019 showed that in the absence of visual cues OV cell tuning degrades but does not disappear (FigS7d-e), which could potentially be due to the presence of tactile information in the dark. In the present study, tactile cues have been minimized, but the detection of different textures (wall vs transparent film or vs adhesive tape) by the mice cannot be excluded. Thus, other*

fundamental features of objects might also be able to elicit OV cells firing. I do not require the authors to conduct further analyses but they should mention this potential confound in their discussion.

This is an excellent suggestion. We have now dedicated one paragraph of the Discussion to this argument, essentially given as above, but generalized to include both tactile- and olfactory stimuli. That is, although a visual cue clearly is sufficient to elevate OV activity, we cannot exclude that a simple tactile or olfactory cue also could be sufficient.

“However, the results do not rule out that other types of sensory stimuli – such as tactile or olfactory stimuli – also could elevate OV activity. A residual OV response is present even in complete darkness²³, which might result from tactile or olfactory information. Similarly, the amount of tactile information for the visual contrasts was minimised but not absent (for example, a faint edge between the wall and the visual contrast was present). Consequently, while we have identified a fundamental feature that drives OV activity, future work will have to determine whether other sensory features of objects, such as tactile or olfactory cues, also could elicit vectorial responses.”

3. Responsiveness to low contrast:

The authors claim that “a visual contrast is sufficient for OV cells to respond, with increasing certainty for higher contrasts.” The authors should discuss if the variability in OV responses to low contrast stimuli is a direct encoding of the perceptual quality of the object or if it might simply reflect a failure of the mice to detect the visual contrast at some time points. One resolution would be if the authors had simultaneously recorded cells responding and cells not responding to a low contrast object (this would exclude the second scenario).

This is an interesting question, with implications that extend to other functional cell types. For example, a grid cell is more likely to spike when the animal is inside one of its grid fields – but not certain to spike. In that case too we might wonder, what is the source of the variability? Perhaps the animal was not engaged in spatial navigation at the time (analogous to a failure of the mouse to detect the visual contrast) or perhaps the cell was genuinely not responding to that part of space at the time (analogous to encoding of the perceptual quality).

The idea to look at simultaneously recorded cells, in order to weigh the plausibility of the above scenarios, is a good one. Because all recordings were made with tetrodes, we do not typically have many simultaneously recorded OV cells. We do have one pair of simultaneously recorded OV cells for the low-contrast stimulus. Inspection of these data revealed that one cell produces an obvious response while the other cell fails to respond. This would argue in favour of scenario 1, where the source of the variability might be from the individual cell rather than the animal’s perception. However, we feel that one pair of cells is not sufficient to draw any conclusions and have therefore chosen not to conclude on this issue. Instead, we add the caveat that inattention may contribute to weak responses to low-contrast stimuli by noting the following in the Discussion:

“In the absence of a population of cells where some cells respond to low-contrast stimuli and others do not, we cannot rule out, however, the possibility that moments of inattention contribute to the weak response to these stimuli in many cells. .”

4. L246-250 *“That is, spatial representation in MEC might be functionally separated into two distinct systems: one involving grid cells, speed cells and HD cells (dependent on PV interneurons via grid cells and speed cells³⁵) for self-motion-dependent representation in open spaces^{2,36} and one involving OV cells and border cells (putatively dependent on SOM interneurons via OV cells) for more landmark-based navigation. “*

Whereas such a dichotomy is appealing, I cannot see in Miao et al. 2017 elements allowing to support the claim ‘dependent on PV interneurons via grid cells and speed cells³⁵’ and ‘putatively dependent on SOM interneurons via OV cells’. Instead, this study shows that PV or SOM interneuron inactivation has no effect on head-direction cells and border cells. Could the authors make this statement more nuanced or add complementary references?

The Reviewer is correct that Miao et al. 2017 found that neither head direction cells nor border cells were affected by the interneuron manipulations. Instead, only grid cells and speed cells depended on PV interneurons and only aperiodic spatial cells depended on SOM interneurons. The text in brackets “(dependent on PV interneurons via grid cells and speed cells)” and “(putatively dependent on SOM interneurons via OV cells)” was meant to indicate this. However, we see that the paragraph can be easily misinterpreted to imply that all functional cell types in MEC fit neatly into such a scheme. The division between (1) grid cells/speed cells/head direction cells and (2) object-vector cells/border cells is a working hypothesis but not directly suggested by the data from Miao et al. (2017). Perhaps this is also why Reviewer 1 thought that the paragraph was off topic. To avoid confusion, we have rephrased so that we avoid mentioning other cell types than grid cells and OV cells:

“While this has not yet been tested, distinct inhibitory control of OV cells and grid cells might suggest that the two cell types separate into two distinct systems: one involving grid cells (dependent on PV interneurons) for representing the animal’s position by updating self-motion^{2,36} and one involving OV cells (putatively dependent on SOM interneurons) for vectorial representation of the animal’s position by anchoring onto objects and their visual features.”

5. Typos:

Main text

a. *In the following sentences, reference 8 does not seem the correct one (Kropff et al. 2015, Speed cells in the MEC):*

L41 *“In the latter pair of trials, the object was a 40×8×8 cm (height × base area) multi-colour Duplo tower, known to produce clear OV responses⁸ (Supplementary Fig. 2)”*

L46. *As in earlier work⁸, cells had to pass multiple criteria to count as OV cells.*

Yes, the Reviewer is correct. Both sentences should refer to the first object-vector cell paper (Høydal et al. 2019). We have now corrected this mistake.

b. L52. *“the firing fields were displaced > 4 cm away from the object centre.”*
‘Displaced’ is slightly confusing, ‘positioned’ would be more appropriate.

We agree that ‘positioned’ is a better word choice (since ‘displaced’ implies movement of the firing fields) and have changed the sentence accordingly.

c. L126. *“While some cells had firing rates that clearly increased with visual contrast (Fig. 4b) other cells were more difficult to evaluate by eye (Supplementary Fig. 5a)”*
Specify which firing rate (Is it inside ROI? overall FR?).
Referring to evaluation by eye is slightly confusing.

We have specified that this refers to the cells’ firing rate inside the ROI and taken away “by eye” (which indeed was slightly confusing) so that the sentence now reads:

“While some cells had firing rates inside the ROI that clearly increased with visual contrast (Fig. 4b) other cells were more difficult to evaluate (Supplementary Fig. 5a)”

Methods section

L259: buprenorphine, meloxicam: specify why these drugs are injected.

L261: buprenorphine, meloxicam, bupivacaine: specify doses.

We agree that it is important to specify the reasons for using different drugs, as well as their doses, to make it easier to compare surgical procedures across labs. The information about buprenorphine and meloxicam is now updated:

“After induction of anaesthesia, the mice received subcutaneous injections of (a) buprenorphine (Temgesic, 0.03 mg/ml solution and 0.05 mg/kg of animal’s body weight), a centrally acting opioid and (b) meloxicam (Metacam, 1 mg/ml solution and 5 mg/kg of animal’s body weight), a non-steroidal anti-inflammatory drug that inhibits prostaglandin synthesis. The purpose of using these drugs is to reduce postoperative pain (buprenorphine, meloxicam) and to reduce inflammation (meloxicam).”

Related to this, we also specified the dose and purpose of using bupivacaine:

“The local anaesthetic bupivacaine (Marcain, 0.5 mg/ml solution and 1 mg/kg of animal’s body weight) was injected subcutaneously before the incision was made, in order to provide pain relief at the site of the incision.”

Legends

Fig

S8:

- Rationalize the way the number of OV cells are denoted (n in the methods and N in the legend of Fig S8)

We thank the Reviewer for noticing the inconsistent notation. The number of OV cells is now denoted n throughout the manuscript, including in the Supplementary Fig. 8 legend.

*- The authors probably wanted to say 'a firing rate change < 0 Hz' in the following sentence:
"N is the total number of OV cells we have data from, r is the number of OV cells that respond with a firing rate > 0 Hz and $N - r$ is the number of OV cells that respond with a firing rate < 0 Hz."*

That is correct - the sentence above has been changed according to the Reviewer's suggestion.

REVIEWERS' COMMENTS:

Reviewer #1 (Remarks to the Author):

The authors have responded satisfactorily to my comments and I appreciate the additional data added, especially the spike plots. I remain concerned by the small numbers of cells, but accept the argument that the repeated trials constitute multiple replications with each cell. I think however that these results still stand as "preliminary" - but I do think they are interesting and stimulating, and for this reason am supportive of publication.

Reviewer #3 (Remarks to the Author):

The authors satisfactorily addressed my points.